# Liver governs adipose remodelling via extracellular vesicles in response to lipid overload

Yue Zhao [1,2], Meng-Fei Zhao [1], Shan Jiang [1], Jing Wu [1,3], Jia Liu [1], Xian-Wen Yuan [1,2], Di Shen [1], Jing-Zi Zhang [1], Nan Zhou [2], Jian He [2], Lei Fang [1,2]*, Xi-Tai Sun [2]*, Bin Xue[3,4]* & Chao-Jun Li [1,2]*

Lipid overload results in lipid redistribution among metabolic organs such as liver, adipose, and muscle; therefore, the interplay between liver and other organs is important to maintain lipid homeostasis. Here, we show that liver responds to lipid overload first and sends hepatocyte-derived extracellular vesicles (EVs) targeting adipocytes to regulate adipogenesis and lipogenesis. Geranylgeranyl diphosphate synthase (Ggpps) expression in liver is enhanced by lipid overload and regulates EV secretion through Rab27A geranylgeranylation. Consistently, liver-specific *Ggpps* deficient mice have reduced fat adipose deposition. The levels of several EV-derived miRNAs in the plasma of non-alcoholic fatty liver disease (NAFLD) patients are positively correlated with body mass index (BMI), and these miRNAs enhance adipocyte lipid accumulation. Thus, we highlight an inter-organ mechanism whereby the liver senses different metabolic states and sends corresponding signals to remodel adipose tissue to adapt to metabolic changes in response to lipid overload.

---

[1] State Key Laboratory of Pharmaceutical Biotechnology, Medical School of Nanjing University & Model Animal Research Center, Nanjing University, Nanjing 210093, China. [2] MOE Key Laboratory of Model Animals for Disease Study, Department of Hepatobiliary Surgery & Department of Radiology, Nanjing Drum Tower Hospital, The Affiliated Hospital of Nanjing University Medical School, Nanjing 210093, China. [3] Core Laboratory, Sir Run Run Hospital, Nanjing Medical University, Nanjing 211166, China. [4] State Key Laboratory of Natural Medicines, China Pharmaceutical University, Nanjing 210009, China. *email: njfanglei@nju.edu.cn; sunxitai@vip.qq.com; xuebin@njmu.edu.cn; licj@nju.edu.cn

When a high-fat diet (HFD) is consumed for an extended period, triglycerides (TGs) first accumulate in inguinal white adipose tissue (iWAT) and epididymal fat white adipose tissue (eWAT), and liver steatosis occurs as a consequence of ectopic lipid accumulation[1–3]. Communication among metabolic tissues regulates TG distribution in the body, which is critical for whole-body metabolic homeostasis. When lipid storage and clearance equilibrium in these fuel storage organs are abnormal, the body suffers from metabolic disorders such as obesity, diabetes and hyperlipidaemia.

Liver absorbs chylomicrons (CMs) and other non-lipid nutrients from the hepatic sinusoid postprandially via fenestrated, discontinuous endothelium. Dietary fatty acids are converted into TGs and stored as lipid droplets in hepatocytes; excess lipids are exported into the blood and then stored in adipose and other tissues[1,4]. Thus, the liver is a pleiotropic endocrine organ that first accesses nutrients and then affects metabolism in both local and distal organs. Previous work has shown that hepatic fat content is the strongest predictor of insulin resistance[5,6] and normally precedes the development of other abnormalities, including adipocyte hypertrophy and obesity development[7], adipocyte death[7], macrophage infiltration and inflammation of adipose[7–9]. Several hepatokines, such as fetuin A[10], adropin[11] and fibroblast growth factor 21 (FGF21)[12], regulate insulin sensitivity, inflammation and thermogenesis of target organs, including adipose, bone and heart[5,13,14]. Collectively, liver has important roles in the interplay of the metabolic responses among the metabolic organs, as it conveys its energetic requirements to other organs. However, the mechanism by which cell-specific signals from the liver modulate other organs remains unknown.

Recently, extracellular vesicles (EVs) have been shown to display marked potential in the organ-to-organ communication of metabolic signals[15–17], especially exosomes and microvesicles[18]. Obviously, EVs secreted by the liver may represent a specific strategy to direct their cargo proteins or miRNAs to target tissues[19,20], and the cargo inside also functions in a cell-dependent manner[15,21]. Early evidence suggests that EVs released from adipose modulate local or distal metabolic tissues, including endothelial cells, liver, skeletal muscle and pancreas[15,16,22]. EVs produced by the liver regulate tumour growth, viral infection and hepatocyte regeneration[23,24]. In this study, we provide evidence that, in the context of lipid overload, hepatic EVs mediate target communication within adipose, thus contributing to the multifarious cross-talk between liver and adipose.

Here, we report that TGs first accumulate in the liver followed by adipose of mice fed a HFD for a short period. The short-term lipid overload causes hepatic EV-stimulated adipose adipogenesis, whereas long-term lipid overload induces adipose lipogenesis. This phenomenon is inhibited in our liver-specific geranylgeranyl diphosphate synthase (Ggpps) knockout (LKO) mouse model, in which hepatic EV secretion is reduced. This is found to be the result of the tissue-targeting and metabolic signalling function of liver EVs, which is dependent on the specific delivery mechanism directed by GGPPS-mediated Rab27A geranylgeranylation. Moreover, the levels of several miRNAs derived from EVs are elevated in the plasma of NAFLD patients with excess TG deposition in adipose, and this increase corresponds to enhanced adipocyte TG deposition in vitro and in vivo. Our findings suggest that the liver has a crucial, active metabolic regulatory role by directing intertissue cross-talk with adipose via specific EVs containing miRNAs. Overall, this hepatic EV-mediated liver–adipose signalling axis may be critical for the metabolic adaptation of adipose to lipid changes in maintaining systemic homoeostasis.

## Results

**Liver responds to acute lipid overload first in mice.** To examine the time-dependent metabolic responses of metabolic organs in overnourished mice, we performed a detailed time-course observation in mice fed a HFD. Although body weight was similar between control mice and mice fed a HFD within 48 h after initiating dietary changes (Fig. 1a), the liver weight/body mass ratio started to increase by 6 h in HFD mice, but neither the iWAT or eWAT weight/body mass ratios increased until 12 or 24 h in HFD mice (Fig. 1b, c and Supplementary Fig. 1a and b). However, the gastrocnemius muscle weight/body mass ratio demonstrated no significant change after 1 week of HFD consumption (Supplementary Fig. 1c). Notably, the increase in the liver weight/body mass ratio was a transient phenomenon, as it decreased after 12 h of HFD exposure (Fig. 1b). Haematoxylin and eosin (H&E)-stained liver and quantitative measurement of TGs in liver indicated significant TG accumulation in the liver as early as 6 h after HFD exposure (Fig. 1d, e). An analysis of lipid metabolism-related gene expression patterns indicated that the fatty-acid transport genes Cd36, Ldlr and Lrp, the lipogenesis genes Fas, Scd1 and Acc1, and the lipid oxidation genes Acox, Cpt1a and Acadl were significantly upregulated in the liver as early as 3 h after beginning HFD consumption (Fig. 1f). However, TG accumulation and changes in lipid metabolism-related gene expression in adipose tissue lagged behind those in the liver (Fig. 1g–i). Moreover, TG content and lipid metabolism-related gene expression in the gastrocnemius muscle did not significantly change with HFD until after 1 week of consumption (Supplementary Fig. 1d–f). Interestingly, the expression of adipogenesis genes in iWAT increased at 24 h after HFD treatment (Fig. 1k and Supplementary Fig. 1j), whereas the adipocyte number increased at 1 week as measured by DNA of total adipose tissue, mature adipocytes and the stromal vascular fraction (SVF) (Fig. 1j and Supplementary Fig. 1g). However, the adipogenesis gene expression levels and adipocyte number in eWAT showed no significant changes within 1 week after initiating HFD consumption (Fig. 1j, k and Supplementary Fig. 1h). Meanwhile, the adipocyte size in iWAT and eWAT increased at 12 and 24 h, respectively (Supplementary Fig. 1i). The above observations suggested that the liver is the first organ to respond to acute lipid overload in mice. TGs accumulated first in the liver followed by WATs through adipogenesis and lipogenesis. This phenomenon might occur because, from an anatomical perspective, the liver accesses consumed nutrients more easily than adipose tissue or skeletal muscle.

**Hepatocytes remodel adipocytes via EVs after lipid overload.** It is well accepted that the liver produces a series of circulating factors that modulate the functions of other organs. Thus, we challenged differentiating 3T3-L1 preadipocytes with medium from isolated primary hepatocytes of HFD-fed mice. ORO (ORO) staining and gene expression analysis showed that HFD-hepatocyte-conditioned medium (HFD-CM) significantly enhanced TG accumulation in 3T3-L1 preadipocytes (Fig. 2a, b). However, when we assessed the expression of hepatokines and cytokines previously reported to modulate adipose homoeostasis, there were no significant changes in expression of these genes in the livers of HFD-induced mice, except for Tgf1β (Supplementary Fig. 2a). Of particular interest was FGF21, which has a positive effect on adipose remodelling[12,25]. However, Fgf21 mRNA in the liver and FGF21 protein in the plasma of HFD-fed mice as well as conditioned medium from primary hepatocytes isolated from HFD-fed mice remained unchanged compared with those in mice fed a normal chow diet (Supplementary Fig. 2a–c). In addition, cell lysis and apoptosis assessments showed no significant changes between the hepatocytes of mice fed a chow diet and those fed a HFD (Supplementary Fig. 2d and e). Therefore, we speculated that other hepatic secretory factors are responsible for regulating TG deposition in 3T3-L1 preadipocytes.

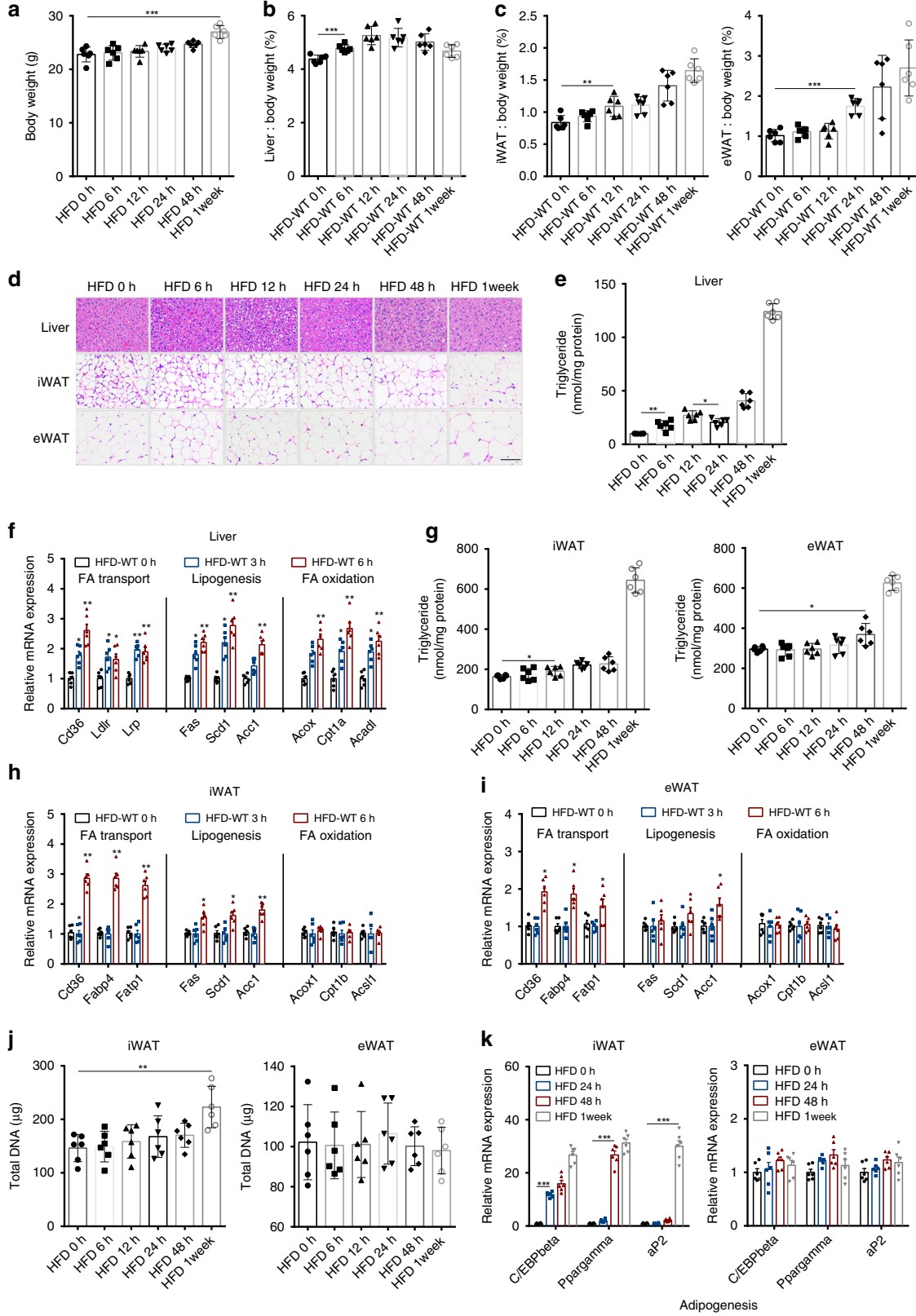

Recent data suggest a novel cell-specific targeting and metabolic signalling role for EVs as natural conveyors of information between cells and across various tissues through the lateral transfer of macromolecules[16,17,21,22,26]. To examine the function of hepatic EVs, we isolated primary hepatocytes from mice fed a normal chow diet or a HFD and then cultured these cells for 48 h. After dead cells and debris were removed from the conditioned medium, electron microscopy (Fig. 2c) and nanoparticle tracking analysis (NTA) (Supplementary Fig. 3f) revealed that microvesicles isolated by $100,000 \times g$ ultracentrifugation

**Fig. 1 The liver responds to acute lipid overload first in mice. a** Body weight of mice. **b–c** Percentages of liver **b**, iWAT **c** and eWAT **c** weight relative to the whole-body weight of HFD-fed mice at different time points. **d** H&E staining of liver, iWAT and eWAT from HFD-fed mice at different time points. (Scale bar: 50 μm). **e** TG content in the liver. **f** Expression of genes related to fatty-acid transport, lipogenesis and fatty-acid oxidation in the liver of HFD-fed mice at the indicated times. **g** TG contents in iWAT and eWAT. **h–i** Expression of genes related to fatty-acid transport, lipogenesis and fatty-acid oxidation in iWAT and eWAT of HFD-fed mice at the indicated times. **j** Quantification of adipocyte number in iWAT and eWAT from HFD-fed mice at different time points. **k** Expression of genes related to adipogensis in the WATs of HFD-fed mice at the indicated times. Six-week-old C57BL/6J mice were fed a HFD for 0 h, 6 h, 12 h, 24 h, 48 h and 1 week ($n = 6$ mice per group). iWAT: inguinal white adipose tissue. eWAT: epididymal white adipose tissue. Data are expressed as the mean ± SEM. *$P < 0.05$, **$P < 0.01$, ***$P < 0.001$, unpaired $t$ test. Source data are provided as a Source Data file. See also Supplementary Fig. 1.

contain an abundant number of hepatic EVs with a diameter of ~150 nm (10–15% underestimated owing to fixation and dehydration) compared with the number in EV-free medium (Supplementary Fig. 3d). Density gradient ultracentrifugation analysis[27] of these EVs confirmed that they primarily consist of EVs at fractions 2 and 3 (F2 and F3, respectively) (Supplementary Fig. 3a and e left), consistent with that published for exosomes (1.13–1.19 g per mL)[28]. The EVs expressed the exosome-specific protein markers TSG101, Syntenin-1 and CD63 and were largely in F2–F3 (Supplementary Fig. 3b). Furthermore, histone 3 and calreticulin were not detectable in the hepatic EV fraction (Supplementary Fig. 3c), indicating that the EVs isolated from primary hepatocytes with this protocol contained little to no cellular debris contamination.

We excluded the contamination of potential non-EV carriers of miR-122 because miR-122 outside of EVs was washed out to the greatest extent by density gradient ultracentrifugation (Supplementary Fig. 3e right). In addition, we found that hepatic miR-122 (a specific miRNA enriched in hepatocytes) was increased in mouse serum EVs and adipose after short-term and long-term HFD consumption (Supplementary Fig. 3g–j). The EV markers CD63, CD81, TSG101, Syntenin-1[27] (Fig. 2d) and overall EV production (Fig. 2e) suggested that hepatic EV abundance was enhanced in response to HFD consumption. Furthermore, EV collected from HFD-CM (HFD-EV) enhanced TG deposition in 3T3-L1 preadipocytes, and a several-fold increase in hepatic miR-122 expression was observed in the adipocytes (Fig. 2f, h and Supplementary Fig. 3k). Importantly, prior addition of GW4869 (an inhibitor of EV secretion) to the hepatocytes blocked EV production, the delivery of miR-122 from hepatocytes into 3T3-L1 adipocytes and lipid accumulation in adipocytes, showing that hepatocytes secrete extracellular miRNAs predominantly in an EV-dependent manner (Fig. 2f, g and Supplementary Fig. 3l). Interestingly, when we challenged fully differentiated 3T3-L1 preadipocytes (8 days, the stage that adipogenesis finishes but lipogenesis proceeds) for 2 days with HFD-EV from mice subjected to HFD consumption for 6 h or 12 weeks, the effects of EV on adipogenesis and lipogenesis were different. The 5-ethynyl-2′-deoxyuridine (EdU)-positive cells showed there is no difference in proliferation between HFD-6h EV-treated cells and HFD-12w EV-treated cells (Supplementary Fig. 3m). Although ORO staining showed no significant change between the two groups (Fig. 2i), the HFD-12w EV still stimulated lipid accumulation by day 10 of methylisobutylxanthine/dexamethasone/insulin (MDI) treatment as indicated by lipogenesis gene expression and lipid staining (Fig. 2k and Supplementary Fig. 3n), whereas the HFD-6h EV no longer stimulated further lipid accumulation by day 10 of MDI (Fig. 2j and Supplementary Fig. 3n), showing the different effect of HFD-6h EV. Taken together, these data indicate that upon exposure to excess lipid, the liver remodels adipose in an EV-dependent manner.

**Liver-driven adipose remodelling is mediated by liver Ggpps.** The formation and release of EVs from a cell involves the transportation, interaction and fusion of endosomal vesicles. We have reported that Ggpps, a key enzyme in the mevalonate pathway for GGPP synthesis, is critical for the formation of the docked granule pools involved in insulin release[29] and endosomal function[30]. We found that HFD consumption enhanced Ggpps expression in mouse livers within 12 h after starting consumption, and FFA treatment directly increased Ggpps levels in primary hepatocytes as soon as 8 h after treatment (Fig. 3a). In addition, hepatocytes with *Ggpps*-knockdown exhibited clustered distribution of CD63-positive vesicles and decreased EV secretion compared with control cells (Fig. 3b, c). We further established a liver-specific *Ggpps* knockout mouse (*Ggpps* LKO, Supplementary Fig. 4a–c) to study hepatic EV secretion and TG deposition in WATs after lipid overload. When we subjected *Ggpps* LKO mice to lipid overload conditions, eWAT and iWAT fat masses significantly decreased, as quantified by nuclear magnetic resonance (Fig. 3d, e). The weights of both iWAT and eWAT deposits were significantly decreased (Fig. 3f, g), and the sizes of both iWAT and eWAT were reduced in *Ggpps* LKO mice (Fig. 3h, i), whereas the cell number (based on DNA content) in the fat tissue in both iWAT and eWAT did not differ between the WT and LKO mice (Supplementary Fig. 4d). Thus, the reduction in adipose mass in *Ggpps* LKO mice was attributable to decreased adipocyte size in both iWAT and eWAT. Meanwhile, hepatic *Ggpps* deletion improved systemic glucose tolerance but not insulin sensitivity (Supplementary Fig. 4e and f). Analyses of lipid metabolism-related gene expression in adipose with lipogenesis and lipolysis assay also indicated that *Ggpps* deletion in the liver significantly decreased lipogenesis and enhanced lipolysis in iWAT (Fig. 3j and Supplementary Fig. 4g and h). These genetic and physiological data suggest that Ggpps in hepatocytes regulates liver-driven WAT remodelling.

**Ggpps regulates EV secretion via Rab27A geranylgeranylation.** We found that *Ggpps* deficiency did not affect cell viability (Supplementary Fig. 5a and b) or hepatokine secretion upon examination of several hepatokines, such as FGF21 in the liver, plasma or primary hepatocytes (Supplementary Fig. 5c–e). We then determined whether EVs mediate liver-driven WAT remodelling by Ggpps. The production of EV secreted from primary hepatocytes isolated from LKO mice was significantly decreased compared with that in hepatocytes isolated from wild-type mice (Fig. 4a). *Ggpps* deficiency inhibited EV secretion as determined by the expression of the EV markers CD63 and TSG101 (Fig. 4b). Regarding endosomal function, prenylation of Rab is indispensable owing to its role in promoting localisation to membranes, where Rab exerts its functions to mediate EV secretion. Protein prenylation regulates various Rab-GTPases, most of which contain C-X-C or C-C motifs (C: cysteine) on the C-terminus[31] and function in EV secretion, including Rab7, Rab5A, Rab5B and Rab27A. Previous work has shown that the formation and release of exosomes from cells are regulated by several Rab-GTPases, particularly Rab27[32]. To illustrate the relationship between Ggpps and EV secretion, we examined the

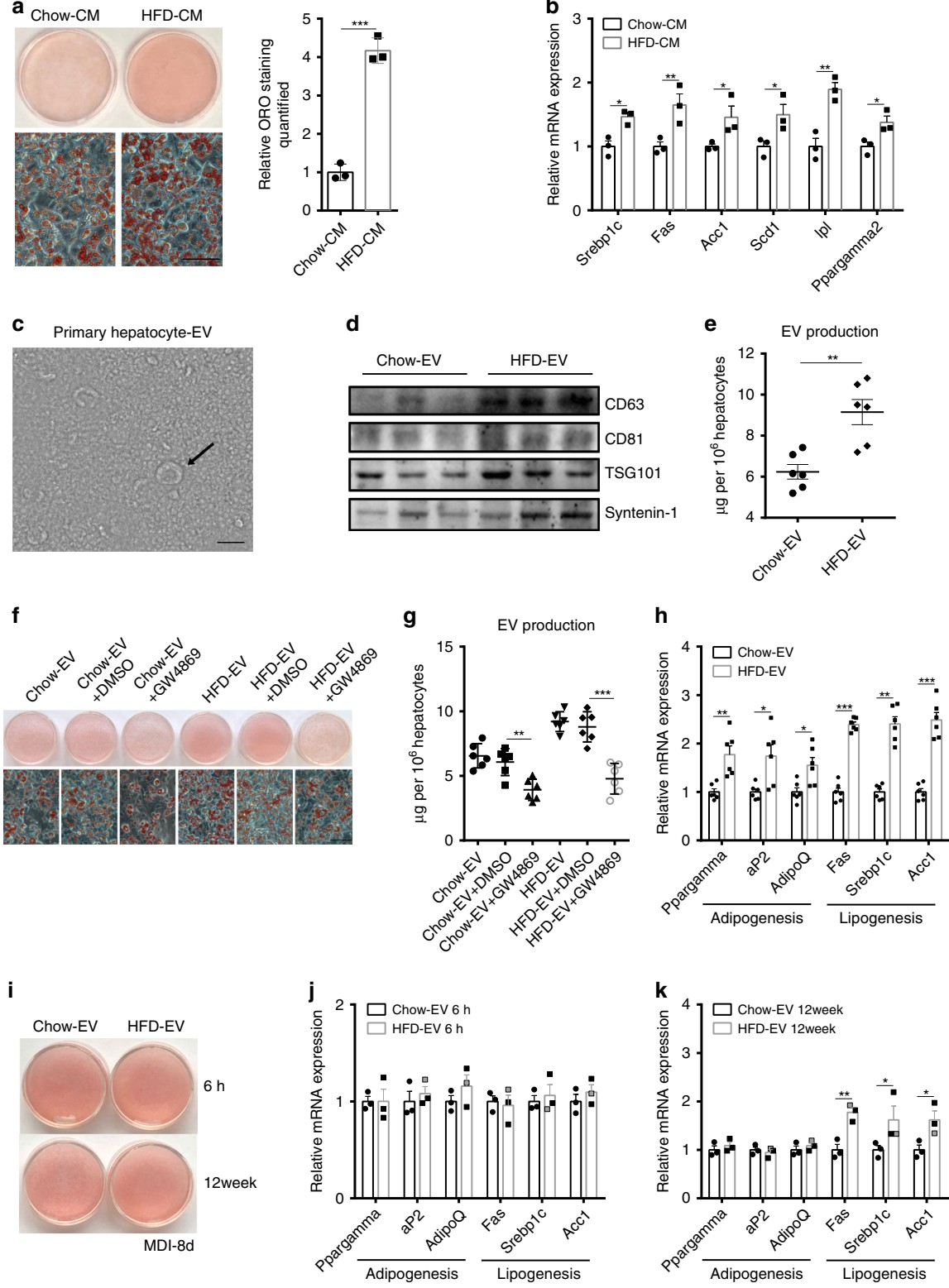

effect of Ggpps on several Rab-GTPases (including Rab27A and Rab5), which regulate the formation and release of EVs. In hepatocytes, geranylgeranylation and the hydrophobic properties of Rab27A (Fig. 4c) and Rab5 (Fig. 4d) were largely increased with lipid overload but were reduced when Ggpps was knocked down. When we transfected primary hepatocytes with plasmids containing wild-type Rab27A or the Rab27AC219S mutant

(mutation of the geranylgeranyl site), we observed that cells expressing Rab27AC219S showed clustered distribution of CD63-positive vesicles and decreased EV production (Fig. 4e–g), and Ggpps overexpression did not reverse the effects of the Rab27A geranylgeranyl site mutation in hepatocytes, suggesting that Ggpps regulates multivesicular endosome distribution and EV secretion in a Rab-GTPase prenylation-dependent manner.

**Fig. 2 Hepatocytes remodel adipocytes via EVs after lipid overload. a** Hepatocyte-conditioned medium (Chow-CM or HFD-CM) prepared from primary hepatocytes from mice fed either a normal chow diet or a HFD that were incubated in EV-free medium for 48 h was added to 3T3-L1 preadipocytes. Adipocytes that differentiated from 3T3-L1 preadipocytes at day 6 after MDI induction were stained with ORO. ORO-positive staining in 3T3-L1 preadipocytes was quantified. ($n = 3$ biologically independent samples, scale bar, 50 μm). **b** Expression of lipogenesis-related genes in 3T3-L1 preadipocytes treated with Chow-CM and HFD-CM. ($n = 3$ biologically independent samples). **c** Electron microscopy analysis of EVs secreted by primary hepatocytes from mice fed a normal chow diet. (Scale bar, 200 nm). **d** The EV-related protein markers CD63, CD81, TSG101 and Syntenin-1 were measured by western blotting. These blots are representative of three independent replicated experiments, each containing three samples. **e** EV production by primary hepatocytes from mice fed a normal chow diet or HFD. ($n = 6$ biologically independent samples). **f** Hepatocyte-derived EVs (Chow-EV or HFD-EV) were added to 3T3-L1 preadipocytes. GW4869 (10 μM) was used to treat donor WT cells to inhibit EV formation. Adipocytes that differentiated from 3T3-L1 adipocytes at day 6 after MDI induction were stained with ORO. (Scale bar, 50 μm). **g** EV production by primary hepatocytes treated with 10 μM GW4869. ($n = 6$ biologically independent samples). **h** Expression of genes related to adipogenesis and lipogenesis in 3T3-L1 preadipocytes from Fig. 2f. ($n = 6$ biologically independent samples). **i** Hepatocyte-derived EVs from mice that consumed a normal chow diet or a HFD for 6 h and 12 weeks were added to 8-day differentiated 3T3-L1 preadipocytes. Adipocytes that differentiated from 3T3-L1 preadipocytes at day 10 after MDI induction were stained with ORO. **j–k** Expression of genes relating to adipogenesis and lipogenesis in 3T3-L1 preadipocytes from Fig. 2i. ($n = 3$ biologically independent samples). Data are presented as the mean ± SEM. *$P < 0.05$, **$P < 0.01$, ***$P < 0.001$, unpaired $t$ test. Source data are provided as a Source Data file. See also Supplementary Figs. 2 and 3.

**Hepatic EVs mediate adipose remodelling by Ggpps.** To verify that hepatocyte-derived EVs can truly enter into adipose in vitro, we labelled EVs with the fluorescent dye PKH67 and then added them to the culture medium of mature adipocytes to visualise any green fluorescence staining in these cells with an Operetta High Content Imaging system for 5 h. We found that the direct regulatory functions in adipocytes mediated by hepatic EVs might be adipocyte-specific, as these EVs directly entered into adipocytes (Fig. 5a and Supplementary Movie 1) but not myocytes (Supplementary Fig. 6a). To further verify that hepatic EVs enter into adipose in vivo, we injected labelled hepatic EVs into WT and LKO mice fed a chow diet or a HFD (Chow-WT, Chow-LKO, HFD-WT and HFD-LKO) via tail vein. We detected a strong hepatic EV signal (red fluorescence) in adipose at 6 h post injection, suggesting that hepatocyte-derived EVs can be taken up by both iWAT and eWAT and that HFD consumption could enhance EVs uptake into adipose tissue. In addition, compared with wild-type hepatocytes, hepatocytes with *Ggpps* deficiency resulted in fewer EVs taken up by adipose tissue, suggesting that liver Ggpps has a critical role in the liver–adipose axis (Fig. 5b and Supplementary Fig. 6b). When we injected mice with either Chow-EVs or HFD-EVs via tail vein, iWAT and eWAT mass and weight (Fig. 5c), lipogenesis-related gene expression (Fig. 5d) and miR-122 levels in adipose (Supplementary Fig. 6c right) were all significantly increased in mice injected with HFD-EV compared with those injected with Chow-EV, whereas the initial weight and food intake of mice showed no significant change (Supplementary Fig. 6c left and middle). In addition, EV collected from the conditioned medium of *Ggpps*-null hepatocytes from HFD-fed mice (HFD-LKO EV) did not enhance TG deposition (Fig. 5e) or lipogenesis-related gene expression (Supplementary Fig. 6d) in 3T3-L1 preadipocytes. Meanwhile, HFD-WT EV increases fat accumulation in the LKO mice, whereas HFD-LKO EV does not (Fig. 5f). A concomitant increase in the expression of lipogenesis genes and decrease in the expression of lipolysis genes in iWAT and eWAT from HFD-LKO mice were also observed after HFD-WT EV treatment (Fig. 5g, h). Moreover, after injecting mice with HFD-WT EV via tail vein, we found that the expression of genes related to lipid metabolism changed rapidly within 30 min after injection in iWAT and 1 h in eWAT (Supplementary Fig. 6e), suggesting that hepatic EVs regulate lipid metabolism-related genes to affect adipocyte function. These results suggest that hepatic EVs mediate liver-driven adipose remodelling through Ggpps.

**Characterisation of miRNAs inside hepatocyte-derived EVs.** Secreted miRNAs have been reported to convey regulatory signals across various tissues[15,22]. Thus, we compared differences in miRNA content in hepatic EVs between chow diet- and HFD-fed mice. The miRNA array showed that HFD-EV exhibited significant alterations in the miRNA expression pattern compared with that in Chow-EV (Supplementary Fig. 7a). Among these altered miRNAs, let-7e-5p had the greatest increase in HFD-EV (Fig. 6a). Our results showed that primary hepatocytes from mice fed a HFD expressed higher intracellular levels of miR-122, let-7e-5p, miR-31-5p and miR-210-3p, leading to increased levels of these miRNAs in EVs (Supplementary Fig. 7b). Consistent with the previous finding that hepatic EVs directly enter adipocytes, we detected increased miR-122, let-7e-5p, miR-31-5p and miR-210-3p levels in adipocytes with endogenous knockout of let-7e-5p (let-7e-5p$^{-/-}$), miR-210-3p (miR-210-3p$^{-/-}$) and miR-31-5p (miR-31-5p$^{-/-}$) when cells were treated with WT EVs derived from primary hepatocytes (Fig. 6b). These miRNAs from EVs increased TG deposition in 3T3-L1 preadipocytes, as EVs from let-7e-5p$^{-/-}$ and miR-210-3p$^{-/-}$ hepatocytes from HFD-fed mice significantly blocked lipid accumulation, whereas EVs from miR-31-5p$^{-/-}$ hepatocytes had no obvious effect as determined by ORO staining (Fig. 6c). However, EVs from let-7e-5p$^{-/-}$, miR-210-3p$^{-/-}$ and miR-31-5p$^{-/-}$ HFD-hepatocytes downregulated the expression of genes related to adipogenesis and lipogenesis (Fig. 6d) and upregulated the expression genes related to fatty-acid oxidation and thermogenesis (Supplementary Fig. 7c).

Meanwhile, a miRNA array and qPCR confirmed that let-7e-5p, miR-210-3p and miR-31-5p levels in EVs were decreased in *Ggpps* LKO mice (Supplementary Fig. 7a and d). The functional loss of EV-driven WAT remodelling was rescued by EVs from hepatocytes transfected with either let-7e-5p mimics or miR-210-3p mimics under HFD conditions (Fig. 6e), which is in accordance with previous findings that miRNA-210-3p regulates adipocyte differentiation through Wnt signalling[33]. Treatment with let-7e-5p mimics enhanced the expression of genes related to adipogenesis and lipogenesis (Fig. 6f) and diminished the expression of genes related to mitochondrial biogenesis (Supplementary Fig. 7e). Extracellular flux analysis further indicated that the mitochondrial respiration of lipids in adipocytes was substantially inhibited by EV-let-7e-5p-mimic (Supplementary Fig. 7f). In addition, the number of mitochondria in adipocytes was decreased by EV-let-7e-5p-mimic (Supplementary Fig. 7g). All these data indicated that EVs containing let-7e-5p increase lipogenesis and reduce lipid oxidation.

To determine whether the transferred let-7e-5p was indeed associated with EVs, we transfected WT cells with Cy3-labelled let-7e-5p mimics, isolated the EVs and then labelled the EVs with a green fluorescent marker. The EVs were then incubated with

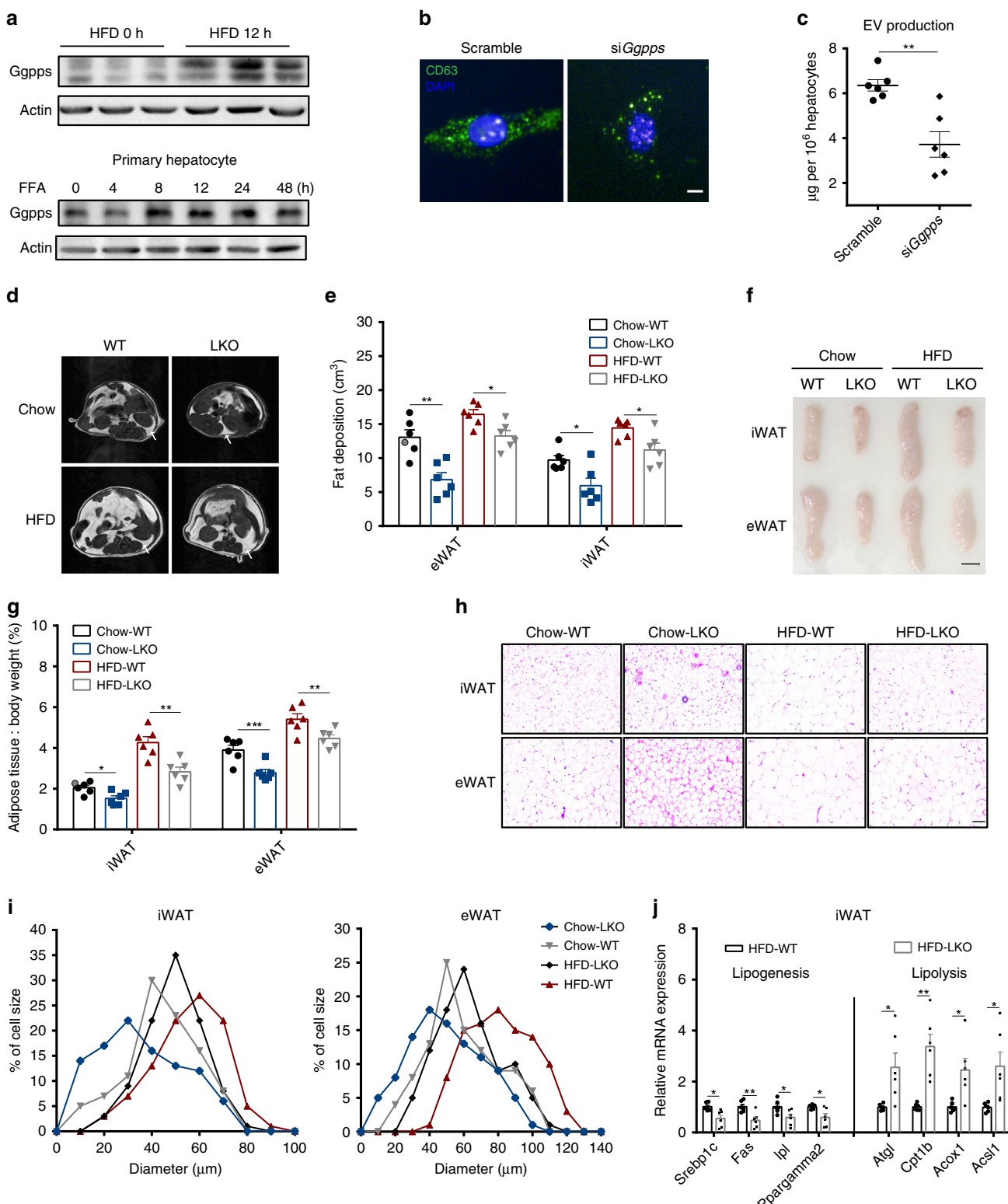

3T3-L1-let-7e-5p$^{-/-}$ cells. After a 5-h incubation with the labelled EVs, Cy3 fluorescence was observed in the recipient cells, in which it was colocalised with the green fluorescence-labelled EV membranes (Fig. 6g), suggesting that let-7e-5p was packaged in the EVs. Together, these data showed that let-7e-5p can be transferred from hepatocytes to adipocytes by EVs.

Based on computational algorithm analysis, the target genes of let-7e-5p are involved in mitochondrial function and

thermogenesis, including PPARγ coactivator-1 α (Pgc1α), Fastk and Acsl6 and Elovl4, the expression of which was decreased by EVs from hepatocytes transfected with let-7e-5p mimics (Fig. 6h). In accordance with this finding, the expression profile of these genes in the white adipose tissue of mice fed a HFD was decreased compared with that in mice fed a chow diet (https://www.ncbi.nlm.nih.gov/geo/query/acc.cgi?acc=GSE14363). Among these genes, Pgc1α showed the most significant change. Moreover,

**Fig. 3 Liver-driven adipose remodelling is mediated by liver Ggpps. a** Ggpps protein levels in livers from HFD-fed (12 h) C57BL/6J mice following the time-dependent FFA incubation of primary hepatocytes. The FFA mixture included oleic acid and palmitic acid at a ratio of 2:1 dissolved in BSA. ($n = 3$ mice per group). **b** Fluorescence microscopy of primary hepatocytes treated with a scrambled control (Scramble) or a Ggpps siRNA (siGgpps). (Scale bar, 5 μm). **c** EV production by WT and Ggpps-knockdown primary hepatocytes. ($n = 6$ biologically independent samples). **d** Whole-body fat distribution in WT and liver-specific Ggpps knockout mice (LKO) fed a normal chow diet or a HFD, as visualised by NMR. **e** Calculation of eWAT and iWAT mass in the mice from **d**. ($n = 6$ mice per group). **f** Comparison of eWAT and iWAT mass in WT and LKO mice fed a normal chow diet or a HFD. ($n = 6$ mice per group, scale bar, 5 mm). **g** Fat index (percentage of fat pad weight relative to the whole-body weight) of iWAT and eWAT in the mice from **f**. ($n = 6$ mice per group). **h** H&E staining of iWAT and eWAT from mice in **g** (Scale bar, 50 μm). **i** Quantification of the diameters of adipocytes from iWAT and eWAT collected from WT and LKO mice fed a normal chow diet or a HFD. Data were collected from H&E-stained sections from three individual mice, five fields per mouse, 10–15 cells per field in each group and analysed using ImageJ software. **j** Expression of genes related to lipogenesis and lipolysis in iWAT from WT and LKO mice fed a HFD. ($n = 6$ mice per group). Data from WT and LKO mice fed a normal chow diet and HFD for 12 weeks (starting at 8 weeks old) are presented as the means ± SEM. *$P < 0.05$, **$P < 0.01$, ***$P < 0.001$, unpaired $t$ test. Source data are provided as a Source Data file. See also Supplementary Fig. 4.

Ggpps deficiency in the liver resulted in decreased let-7e-5p levels and elevated Pgc1α abundance in iWAT and eWAT (Supplementary Fig. 7h and i). Meanwhile, the expression of let-7e-5p changed rapidly in WATs when we injected mice with HFD-WT hepatic EVs via tail vein (Supplementary Fig. 7j). To confirm that hepatic EVs affect adipocyte function through let-7e-5p and Pgc1α, we identified a conserved target site in the 3′-UTR of the Pgc1α mRNA transcript that can bind both human and mouse let-7e-5p. A luciferase assay with reporter plasmids containing the Pgc1α 3′-UTR in 3T3-L1 preadipocytes indicated that let-7e-5p markedly decreased luciferase activity with wild-type Pgc1α 3′-UTR but not with the mutant 3′-UTR site (Fig. 6i). The EV-let-7e-5p mimic repressed mRNA and protein levels of Pgc1α (Fig. 6j), whereas EV-let-7e-5p$^{-/-}$ effectively increased those of Pgc1α (Fig. 6k). These data suggest that let-7e-5p directly targets Pgc1α to affect adipose remodelling.

Pgc1α is an inducible transcriptional coactivator for the biogenesis of mitochondria and is involved in glucose and fatty-acid metabolism for adaptive energy metabolism in adipose and other tissues[34,35]. To further confirm the role of EVs containing let-7e-5p in liver-driven adipose remodelling, we injected HFD-WT EVs transfected with anti-let-7e-5p (anti-let-7e-5p-EV) into the iWAT of mice. Anti-let-7e-5p-EV significantly decreased iWAT mass compared with the mass in mice treated with anti-scramble-EV; this decrease was rescued by decreasing Pgc1α levels with adeno-Pgc1α-shRNA (Supplementary Fig. 7k). The expression of lipid oxidation genes also indicated that anti-let-7e-5p-EV increased the expression of Cpt1b, Acox1 and Mcad, and this increase was reversed by the injection of adeno-Pgc1α-shRNA (Supplementary Fig. 7l), further suggesting that EVs containing let-7e-5p are responsible for liver-driven adipose remodelling.

Reports have shown that miRNAs must be associated with the RNA-induced silencing complex containing the Argonaute (Ago) endonuclease to achieve silencing of their mRNA targets[36]. To analyse whether Ago interacts with hepatic EV-let-7e-5p to directly target Pgc1α in let-7e-5p$^{-/-}$ adipocytes, RNA–protein complexes were immunoprecipitated with antibodies specific to Ago (Supplementary Fig. 8a), and RNA was isolated from the complex for analysis. qPCR analysis showed that treatment of adipocytes with hepatic EVs overexpressing let-7e-5p led to an increase amount of let-7e-5p (Supplementary Fig. 8b) and Pgc1α (Supplementary Fig. 8c) bound to the Ago protein. These observations suggest that by complexing with Ago, hepatic EV-let-7e-5p was transferred to adipocytes, which then targets Pgc1α to regulate lipid deposition.

An analysis of liver samples from patients with steatosis (Supplementary Table 1) also suggested that Ggpps expression levels were directly correlated with TG content in the livers of 23 patients (Fig. 7a). Let-7e-5p was also enriched in the plasma EVs of NAFLD patients with a BMI > 30 compared with its expression in NAFLD patients with a BMI < 30 (Fig. 7l). Although general enrichment of let-7e-5p and the other four miRNAs was observed in the plasma EVs of HFD-fed mice (Fig. 7b–f), only EV-let-7e-5p and EV-miR-210-3p exhibited significant positive correlations with BMI (Fig. 7g–j). We also found that let-7e-5p was more highly expressed than either miR-31-5p or miR-210-3p in liver samples from humans (Fig. 7k). Interestingly, EV-miRNA-210-3p also demonstrated higher expression levels in NAFLD patients with BMI > 30, similar to that observed with let-7e-5p (Fig. 7m–o). These data indicate that miRNAs in hepatic EVs are responsible for liver-driven adipose remodelling in humans. The other factors within EVs that are responsible for this remodelling must be further explored.

Our study suggests that Ggpps in the liver responds to acute and chronic lipid overload and remodels lipid deposition in adipose tissue via adipogenesis and lipogenesis through hepatocyte-derived EVs containing miRNAs such as let-7e-5p (Fig. 8).

## Discussion

Although metabolic organs such as the liver, adipose and skeletal muscle were previously reported to be crucial for regulating lipid homoeostasis, particularly in cases of obesity, exercise and fasting; little is known about which organs play a central role in this communication system under specific physiological conditions. Here, we show that the liver has an early role in the aetiology of metabolic dysfunction associated with lipid overload and then sends a signal to modulate the metabolic functions of adipose, including lipid deposition. This signal is conveyed by hepatic EVs containing miRNAs, which are increased in response to lipid overload and directly target adipocytes by EVs. Among these EV-containing miRNAs, let-7e-5p enhances adipocyte lipid deposition by increasing lipogenesis and inhibiting lipid oxidation through Pgc1α. In summary, these findings indicate that the liver, as a 'sensor' of lipid overload and a 'commander' of lipid metabolism, triggers an integrated response to modulate the metabolic functions of adipose by directly secreting miRNA-containing hepatocyte-derived EVs.

Upon consumption of a normal diet, the liver either takes up dietary glucose and circulating chylomicron remnants and stores them by converting these molecules to TG or packages lipids into lipid-rich lipoproteins to be used by peripheral tissues such as adipose and skeletal muscle. However, long-term nutrient overload, such as increased dietary fat consumption, may exacerbate these phenotypes and prompt TG accumulation in adipose and ectopic lipid accumulation in the liver, causing hepatic steatosis[37,38]; this situation is usually observed when obesity has already developed. However, the response is different at the beginning when the body is first exposed to lipid overload. We noted a discrepancy in fat deposition within the liver, adipose and skeletal muscle after short-term HFD consumption. The liver

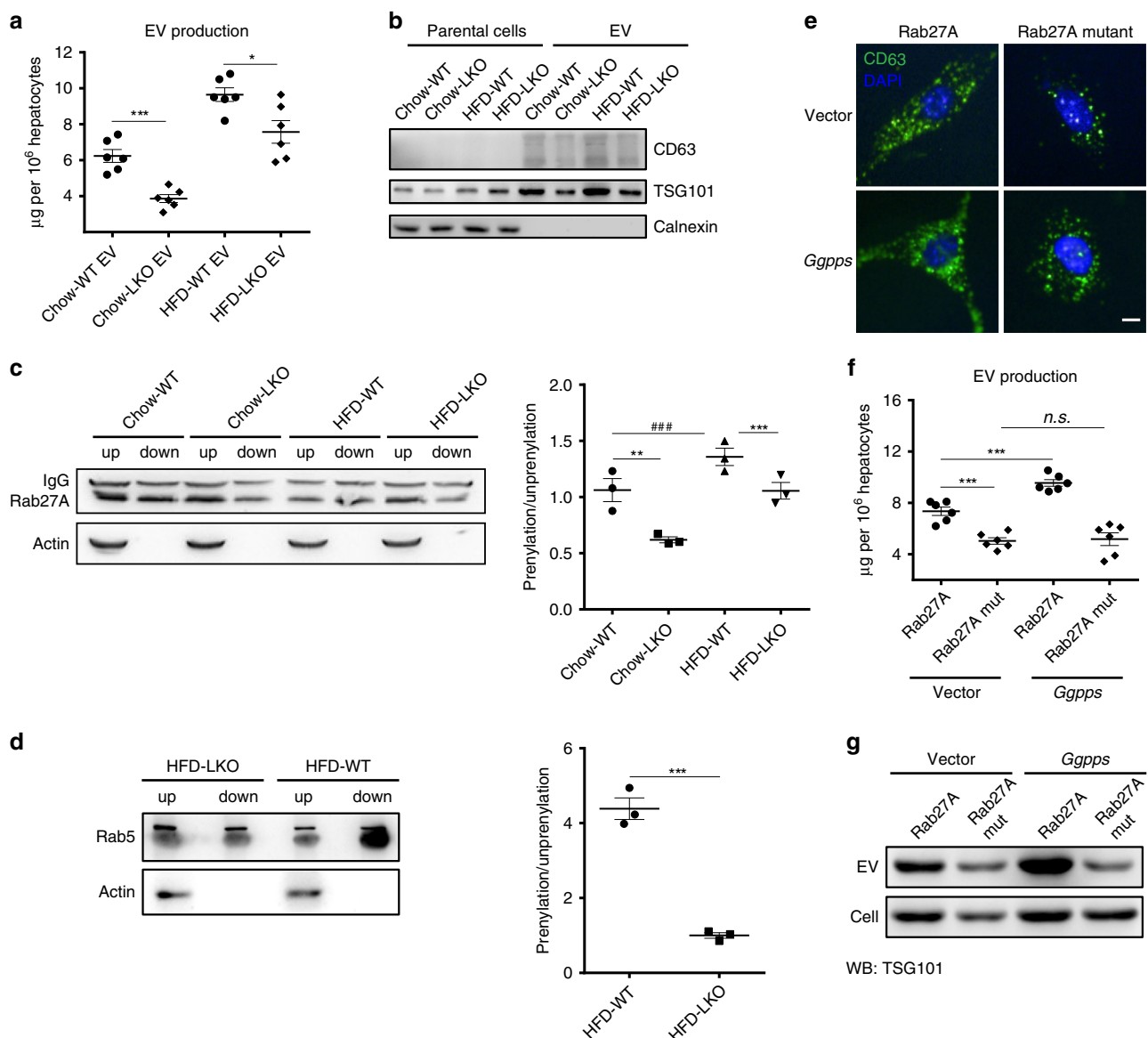

**Fig. 4 Ggpps regulates hepatocyte-derived EV secretion. a** EV production by primary hepatocytes from WT and liver-specific *Ggpps* knockout mice (LKO) fed a normal chow diet or a HFD. ($n = 6$ mice per group). **b** CD63, TSG101 and Calnexin protein levels in parental cells and EVs from WT and LKO mice fed a normal chow diet or a HFD. **c** Membrane-associated Rab27A (lipid-soluble protein with geranylgeranylation in the detergent phase, down) and cytoplasm-associated Rab27A (water-soluble protein with no geranylgeranylation in the aqueous phase, up) were obtained by Triton X-114 extraction and analysed by immunoblotting. ($n = 3$ biologically independent samples). **d** Membrane-associated Rab5 (lipid-soluble protein with geranylgeranylation in the detergent phase, down) and cytoplasm-associated Rab5 (water-soluble protein with no geranylgeranylation in the aqueous phase, up) were obtained by Triton X-114 extraction and analysed by immunoblotting. ($n = 3$ biologically independent samples). **e** Fluorescence microscopy of primary hepatocytes co-infected with empty vector or Ggpps vector and WT or mutant Rab27A and stained with anti-CD63 antibodies and DAPI (nucleus, blue). (Scale bar, 5 μm). **f** EV production by primary hepatocytes from **e**. ($n = 6$ biologically independent samples). **g** TSG101 protein levels in the parental cells and EVs from **e**. Data from 8-week-old WT and LKO mice fed a normal chow diet or a HFD for 12 weeks are presented as the means ± SEM. Groups were analysed using an unpaired *t* test, *$P < 0.05$, **$P < 0.01$, ***$P < 0.001$, WT vs. LKO; ###$P < 0.001$, Chow vs. HFD. Source data are provided as a Source Data file. See also Supplementary Fig. 5.

responds first to lipid overload, followed by adipose. Thus, we speculate that the accumulation of lipids in adipose might be attributable to 'remote control command' by the liver. We found that this command signal is conveyed via hepatic EVs containing miRNAs, including let-7e-5p and miR-210-3p, which target adipocyte mRNA to drive TG redistribution upon lipid overload. Interestingly, we found that HFD-6h EVs no longer stimulated further lipid deposition of differentiated 3T3-L1 preadipocytes by day 8, whereas the HFD-12w EVs still enhanced lipogenesis (Supplementary Fig. 2i–k). Our data indicate that in response to

short-term lipid overload, hepatic EVs mediate adipose function via adipogenesis, leading to a much healthier systemic metabolism to relieve the pressure induced by acute lipid overload, which is in accordance with the phenotype that the adipocyte number and expression of adipogenesis-related genes increased in iWAT after 1 week of HFD consumption (Fig. 1j, k and Supplementary Fig. 1g). The fat-cell progenitors from different fat deposits in the body have distinct properties, so the inherent differences in SVF cell dynamics and the plasticity of iWAT and eWAT may contribute to the distinct responses of different fat deposits to lipid

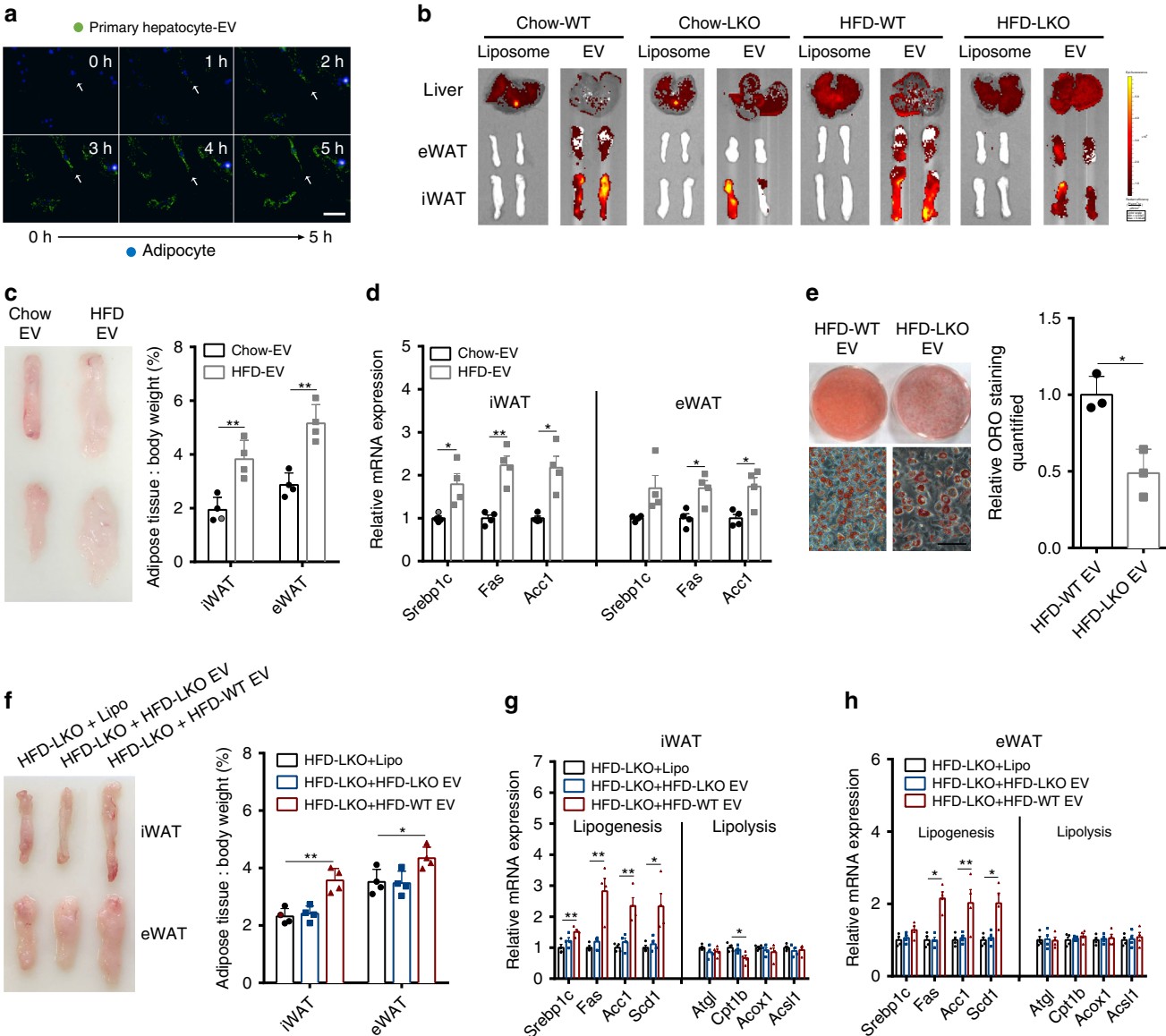

**Fig. 5 Hepatic EVs mediate adipose remodelling by Ggpps. a** Primary hepatocyte-derived EVs uptake in 3T3-L1 adipocytes after 5 h of co-culture. (Scale bar, 50 μm). **b** EVs produced by primary hepatocytes isolated from WT mice fed a normal chow diet were injected via tail vein into WT and LKO mice fed a normal chow diet or a HFD. Empty liposomes were used as controls. Red fluorescence levels in the liver, iWAT and eWAT were acquired after 6 h with an IVIS Spectrum system. **c** EVs produced by primary hepatocytes isolated from mice fed a normal chow diet (Chow-EV) or a HFD (HFD-EV) were injected via tail vein every 3 days for 6 weeks into 8-week-old mice fed a normal chow diet. Comparison of iWAT and eWAT in the mice after treatment with EVs. The percentages of iWAT and eWAT weight relative to the whole-body weight of mice treated with Chow-EV and HFD-EV were calculated. (n = 4 mice per group). **d** Expression of genes related to lipogenesis in iWAT and eWAT of mice treated with Chow-EV and HFD-EV in **c**. (n = 4 mice per group). **e** Hepatic EVs produced from WT and LKO mice fed a HFD (HFD-WT EV and HFD-LKO EV, respectively) were added to 3T3-L1 preadipocytes. Adipocytes that differentiated from 3T3-L1 preadipocytes at day 6 after MDI induction were stained with ORO. ORO-positive areas of 3T3-L1 preadipocytes were quantified. (n = 3 biologically independent samples per group, scale bar, 50 μm). **f** EVs produced by primary hepatocytes isolated from LKO or WT mice fed a HFD (HFD-LKO EV or HFD-WT EV) were injected via tail vein every 3 days for 3 weeks into 8-week-old HFD-LKO mice. The mass of iWAT and eWAT in HFD-LKO mice was compared, as was the fat index of iWAT and eWAT. Empty liposomes were used as controls. (n = 4 mice per group). **g–h** Expression of genes related to lipogenesis and lipolysis in iWAT **g** and eWAT **h** of the mice from **f**. (n = 4 mice per group). Data from 8-week-old WT and LKO mice are presented as the means ± SEM. *P < 0.05, **P < 0.01, unpaired t test. Source data are provided as a Source Data file. See also Supplementary Fig. 6 and Supplementary Movie 1.

overload. The discrepancy in adipocyte number of iWAT and eWAT is in accordance with previous clinical data that fat-cell number increases in certain deposits in adults after only 8 weeks of increased food intake[39]. Meanwhile, long-term lipid overload causes hepatic EV-stimulated adipose hypertrophy via lipogenesis, resulting in a detrimental metabolic phenotype. Therefore,

we found that the adipocyte size increased by 12 weeks HFD, and blocking Ggpps-dependent EV secretion was beneficial by reducing the adipocyte mass (Fig. 3) and improving systemic glucose tolerance. The different effects of hepatic EVs on adipogenesis and lipogenesis in adipocytes may result from the altered 'remote command' of the liver, different EVs, evoked by acute or chronic

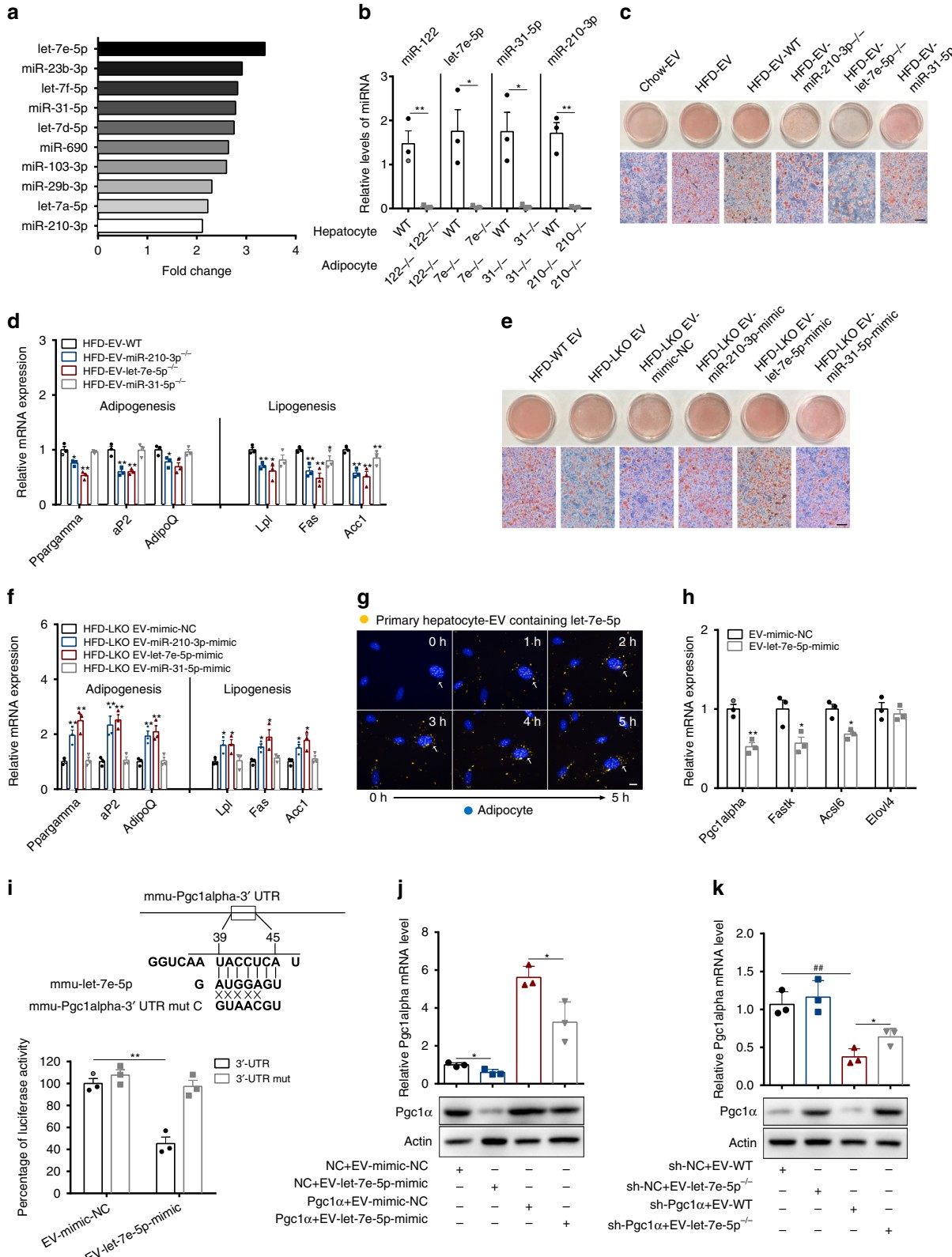

lipid overload. In summary, our work illuminates the differentiation potential of adipose via exposure to hepatic EVs to promote healthier systemic metabolism.

Organ interplay via certain metabolic stimulators is an important characteristic feature of cellular metabolic networks that primarily affects energy redistribution to maintain homeostasis. Liver and adipose influences each other by several

hepatokines and adipokines, like acylcarnitine[40], IL-6[41] and adipocyte-derived exosomes[15]. EVs secreted from macrophages or stem cells that act on adipose tissue can also alter whole-body metabolism[17,22,26]. EV trafficking between endothelial cells and adipocytes is regulated by the systemic metabolic state[16]. However, we still do not know how these tissues sense stress and then orchestrate inter-tissue communication.

**Fig. 6 MiRNAs in hepatic EVs mediate liver-driven adipose remodelling. a** The fold change of several miRNAs under HFD-EVs compared with Chow-EVs. **b** The relative miRNAs levels in miRNA knockout adipocytes that received WT, miR-122$^{-/-}$, let-7e-5p$^{-/-}$, miR-31-5p$^{-/-}$ or miR-210-3p$^{-/-}$ EVs derived from primary hepatocytes. **c** EVs from primary hepatocytes isolated from Chow- or HFD-fed WT mice and transfected with miRNA inhibitors (Chow-EV, HFD-EV, HFD-EV-WT, HFD-EV-miR-210-3p$^{-/-}$, HFD-EV-let-7e-5p$^{-/-}$, HFD-EV-miR-31-5p$^{-/-}$) were added to 3T3-L1 preadipocytes. Adipocytes that differentiated from 3T3-L1 preadipocytes at day 8 after MDI induction were stained with ORO. (Scale bar, 50 μm). **d** Expression of genes related to adipogenesis and lipogenesis in the 3T3-L1 preadipocytes from **c**. **e** EVs from primary hepatocytes isolated from HFD-fed WT or LKO mice and transfected with various miRNA mimics (HFD-WT EV, HFD-LKO EV, HFD-LKO EV-mimic-NC, HFD-LKO EV-miR-210-3p-mimic, HFD-LKO EV-let-7e-5p-mimic, HFD-LKO EV-miR-31-5p-mimic) were added to 3T3-L1 preadipocytes. Adipocytes that differentiated from 3T3-L1 preadipocytes at day 8 after MDI induction were stained with ORO. (Scale bar, 50 μm). **f** Expression of genes related to adipogenesis and lipogenesis in the 3T3-L1 preadipocytes from **e**. **g** Purified PKH67-labelled EVs secreted by primary hepatocytes transfected with Cy3-labelled let-7e-5p were incubated with 3T3-L1 preadipocytes cultured in a chamber. (Scale bar, 5 μm). **h** Expression of genes related to fatty-acid oxidation and thermogenesis in 3T3-L1 preadipocytes cultured with or without EVs from primary hepatocytes transfected with let-7e-5p mimic. **i** Putative miRNA target sites of let-7e-5p within the 3′-UTR of Pgc1α. Relative luciferase activity in 3T3-L1 cells co-transfected with EV-let-7e-5p-mimic and reporter plasmid constructs containing either the WT or mutated 3′-UTR of Pgc1α. **j**–**k** Pgc1α mRNA and protein levels in 3T3-L1 cells co-transfected with plasmids expressing Pgc1α or a scrambled control (NC) and with EV-mimic-NC or EV-let-7e-5p-mimic **j**, or with adenovirus expressing an shRNA targeting Pgc1α (sh-Pgc1α) or a scrambled control shRNA (sh-NC) and simultaneously transfected with EV-WT or EV-let-7e-5p$^{-/-}$ **k**. Data are presented as the mean ± SEM. $n = 3$ biologically independent samples per group. Groups were analysed using an unpaired $t$ test, *$P < 0.05$, **$P < 0.01$; $^{##}P < 0.01$, sh-NC vs. sh-Pgc1α. Source data are provided as a Source Data file. See also Supplementary Figs. 7 and 8.

In our working model, lipid overload causes upregulation of Ggpps expression in the liver via an unknown mechanism; this leads to adipose remodelling through regulation of EV secretion machinery under different dietary conditions. More interestingly, we have found that adipose takes up less EVs in the liver Ggpps deficiency mice, which deserves further research to unravel molecular mechanism. Ggpps is a branch enzyme in the mevalonate pathway, which plays an important role in regulating glucose homoeostasis and insulin sensitivity[42], maintaining male fertility[43] and facilitating cardiomyocyte growth[44,45]. Our results, together with previous findings, suggest that Ggpps influences not only the metabolic state of specific tissues but also the whole-body metabolic state by EV secretion.

Our results also show that the liver exhibits an adipose-targeting effect via EV-containing miRNAs. It is well established that the effects of EV targeting specifically depend on proteins such as integrins, MHC molecules and cytoskeletal proteins[21,46]. However, the precise mechanism underlying the adipose-targeting effect by liver remains unknown. This effect may also result from the specific targets of miRNAs. Analysis of the differentially expressed miRNAs in our study identified let-7e-5p as a highly expressed candidate of interest because let-7e-5p acts as a differentiation inducer in adipose-derived stem cells[47]. In addition, we predicted and confirmed Pgc1α as a target of let-7e-5p. Given the obvious importance of Pgc1α for adipocyte function[34], it is not surprising that Pgc1α mediates let-7e-5p activity to alter adiposity. Future work will be necessary to reveal other miRNAs, proteins and metabolites within these EVs that cause changes in metabolic homoeostasis.

In conclusion, our study suggests that the liver is not just a passive ectopic deposition organ but rather is able to actively send signals to other organs upon exposure to metabolic stress, such as lipid overload, via secretion of EVs containing miRNAs, which broadens our understanding of the liver as a metabolic node that processes fuel. The central role of the liver in orchestrating lipid homoeostasis among tissues reminds us to focus on the regulatory functions of liver within the realm of obesity.

## Methods

**Human studies.** Human liver tissues and sera were obtained from patients who underwent Roux-en-Y in the Drum Tower Hospital Affiliated with the Medical School of Nanjing University in Nanjing, China. We have complied with all relevant ethical regulations for work with human participants. Informed consent for the human study was provided by all participants, and all study protocols were reviewed and approved by the Ethical Committee of the Drum Tower Hospital Affiliated with the Medical School of Nanjing University. This study was registered in International Clinical Trial Registry Platform (ICTRP), with the clinical trial number NCT03296605 as an observational design. Initially, 88 subjects were enroled for this study. Of these subjects, 60 were excluded for the following reasons: females, viral hepatitis, hemochromatosis, or alcohol consumption (> 20 g/d for females and > 30 g/d for males). Several subjects met two or more of these exclusion criteria. Physical examinations, including heights, weights, WC and BP, were measured in accordance with standard protocol. Blood sample data, including glucose and triglycerides, were measured using Beckman AU5412 auto-analyzer according to the manufacturer's protocol. Steatosis was assessed and scored in a scale of 0–3 to estimate the NAFLD activity score (NAS).

**Animal care and use.** C57BL/6J mice were purchased from the Model Animal Research Center of Nanjing University. Liver-specific Ggpps knockout mice were generated by crossing Ggpps-floxed mice with Mx1-Cre transgenic mice. All mice were maintained on a normal 12 h/12 h light/dark cycle on a normal chow diet (Xietong Bio) or a HFD containing 60% of calories from fat (HFD) (Research Diets). For the HFD model, WT and Ggpps LKO mice were fed a HFD for 12 weeks starting at 8 weeks of age. Mx1-Cre expression can be driven by injection of double-stranded RNA polyinosinic–polycytidylic acid (PIPC), resulting in Cre-mediated recombination confined to the liver. Ggpps knockout was achieved by PIPC injection at 7 weeks of age, and the knockout efficiency was detected 1 week later. The animals were given access to water ad libitum. All mice used in the experiments were male. We have complied with all relevant ethical regulations for animal testing and research. All animal procedures were carried out in accordance with the approval of the Animal Care and Use Committee at the Model Animal Research Center of Nanjing University in Nanjing, China, and used protocols approved by the institutional animal care committee (#CS20).

**Primary hepatocytes and cell lines.** Primary hepatocytes were prepared by collagenase digestion via catheterisation of the inferior vena cava (IVC)[48]. Prior to collagenase infusion, the liver was perfused (3–4 min) via IVC after severing the portal vein. When the colour of the liver changed to light brown, perfusion was continued with collagenase for 2 min. Within 2 min of digestion, the liver was monitored for the appearance of cracking on the surface. Perfusion was stopped immediately and the liver was excised out into ice-chilled Buffer. Cells were filtered through 100 μm nylon filter and centrifuged at 60 × g for 6 min at 4 °C. The cell pellet was washed once in Buffer (without collagenase) by resuspending and centrifuging at 60 × g for 6 min at 4 °C. The cell pellet was then mixed with Percoll and centrifuged at 100 × g for 10 min. Hepatocytes were collected as a pellet and washed once with Buffer (no collagenase) and then cultured in medium containing penicillin, streptomycin and 10% FBS. After overnight incubation, the culture medium was switched to EV-free Dulbecco's Modified Eagle Medium (DMEM) supplemented with antibiotics.

To generate adipocytes, 3T3-L1 preadipocytes (gift from Dr. Qi-Qun Tang, Fudan Univeristy) were differentiated in induction medium (DMEM containing 10% FBS, 0.5 μg per mL insulin, 0.5 μM dexamethasone and 0.25 mM 3-isobutyl-1-methylxanthine) until day 2. Cells were then cultured in DMEM supplemented with 10% FBS and 0.5 μg per mL insulin for 2 days, after which the medium was replaced every other day with DMEM containing 10% FBS for 6–10 days. To induce myotube formation, L6 skeletal myoblasts (gift from Dr. Amira Klip, Hospital for Sick Children, Toronto, ON, Canada) were cultured in MEM supplemented with 4% FBS for 8–10 days. To induce the differentiation of 3T3-L1 preadipocytes, hepatocyte-conditioned medium was diluted 50-fold.

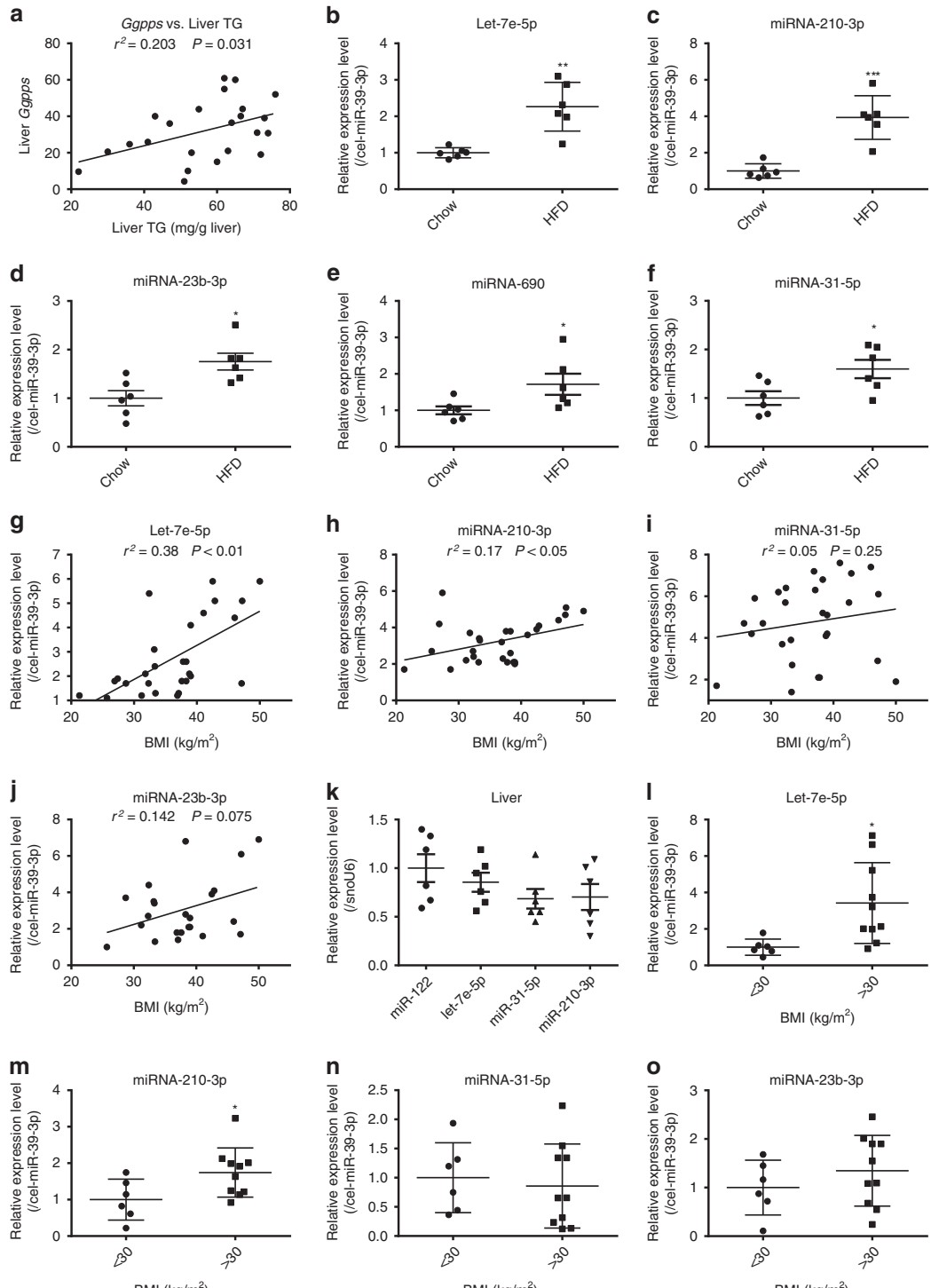

**Fig. 7 *Ggpps* mRNA levels and MiRNA levels of EVs in mice and patients. a** Linear regression analysis between liver TG count and *Ggpps* mRNA levels in patients ($n = 23$ samples). **b**–**f** The levels of let-7e-5p, miR-210-3p, miR-23b-3p, miR-690 and miR-31-5p among the plasma EVs of mice fed a normal chow diet and a HFD. The relative values refer to the fold change of each miRNA in HFD-EVs compared with that in Chow-EVs. ($n = 6$ mice per group). **g**–**j** Linear regression analysis between BMI and let-7e-5p, miR-210-3p, miR-31-5p and miR-23b-3p levels in plasma EVs from patients ($n = 28$ samples). **k** The relative expression levels of miR-122, let-7e-5p, miR-210-3p and miR-31-5p in the livers of patients. The relative values refer to the fold change of let-7e-5p, miR-210-3p and miR-31-5p levels compared to miR-122 levels ($n = 6$ samples). **l**–**o** Let-7e-5p, miR-210-3p, miR-31-5p and miR-23b-3p levels in the plasma EVs of NAFLD patients with a BMI < 30 or > 30. The relative values refer to the fold change of each miRNA in EVs from patients with BMI > 30 patients compared with that in EVs from patients with BMI < 30. (BMI < 30: $n = 6$ samples; BMI > 30: $n = 10$ samples). Data are presented as the mean ± SD. *$P <$ 0.05, **$P <$ 0.01, ***$P <$ 0.001, (**a**, **g**, **h**, **i**, **j**) linear regression analysis; (**b**, **c**, **d**, **e**, **f**, **l**, **m**, **n**, **o**) unpaired *t* test. Source data are provided as a Source Data file. See also Supplementary Table 1.

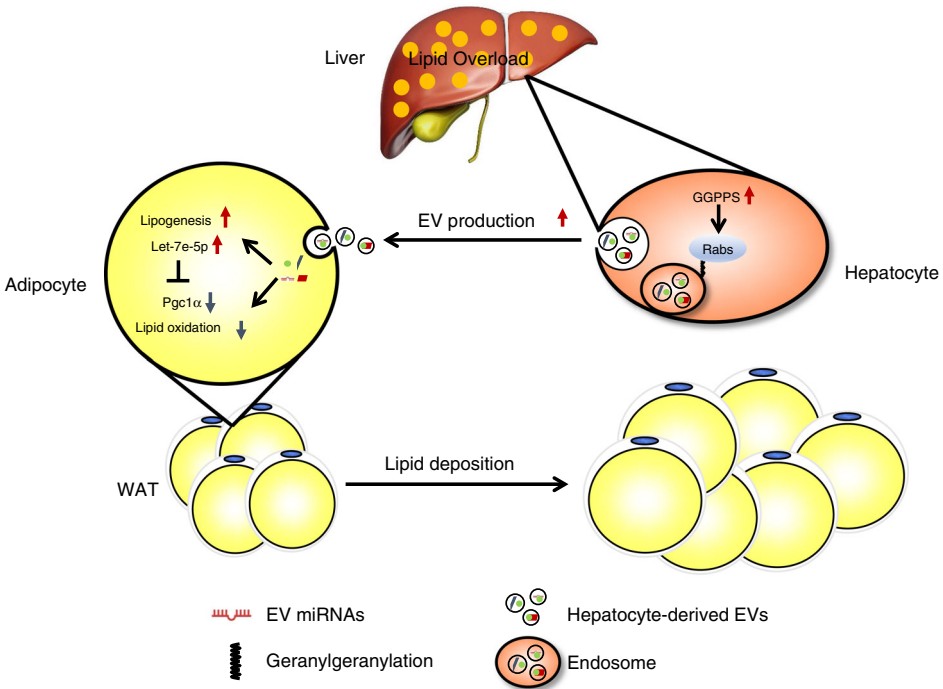

**Fig. 8 Schematic for the contribution of hepatic EVs to adipose remodelling.** Induced Ggpps expression in hepatocytes by acute and chronic HFD consumption gives rise to hepatocyte-derived EV secretion through Rab27A geranylgeranylation, which remodels adipose tissue via adipogenesis and lipogenesis. Hepatocyte-derived EVs containing miRNAs enhance lipid deposition in adipocytes by increasing lipogenesis and inhibiting lipid oxidation through the let-7e-5p-Pgc1α axis.

**Histological analysis**. Standard H&E staining was performed on 5-μm-thick paraffin sections of liver, iWAT, eWAT and Gas muscle. Sections were stained with H&E for structural analysis. For WAT cell size quantification, cell diameters were measured in the H&E-stained sections of three individual samples in each group using ImageJ.

**Quantification of metabolites in mouse tissues and plasma**. TGs (K622-100, BioVision, USA) were measured using colorimetric assays on a NanoDrop 2000c according to the manufacturer's instructions. Plasma concentrations of FGF21 (MF2100, R&D Systems, USA) were measured by enzyme-linked immunosorbent assay using an iMark Microplate Absorbance Reader according to the manu-facturer's instructions.

**Computed tomography**. Six-week-old male C57BL/6J mice fed a HFD for 0 h, 6 h, 12 h, 24 h, 48 h and 1 week were anaesthetised with isoflurane (DS Pharma Animal Health, USA) and analysed using the in vivo micro X-ray CT system R_mCT2 (Rigaku, Japan).

**Mature adipocytes and SVF isolation from adipose tissue**. Adipose tissue was harvested and the SVF cells were isolated by enzymatic digestion (collagenase I; Sigma). The digested tissue was filtered through a 100-μm mesh filter to remove debris. The mature adipocytes floated above the supernatant after centrifugation. The cellular pellet involving the SVF was resuspended with an ammonium chloride lysis buffer to remove red blood cells.

**ORO staining and quantification**. Cells were washed three times with phosphate-buffered saline (PBS) and then fixed for 20 min with 3.7% formaldehyde. Oil Red O (ORO; 0.5% in isopropanol) was diluted with water (3:2) and incubated with fixed cells for 2 h at room temperature. Cells were washed with water, and the stained fat droplets in the cells were visualised by light microscopy and photographed. After the plates were examined under a microscope, they were treated with 100% iso-propyl alcohol to extract the resulting ORO solution, which was then measured at an absorbance of 520 nm.

**LDH release and apoptosis assays**. LDH release was measured using the Cyto-Tox 96 Non-Radioactive Cytotoxicity Assay kit (Promega Corporation, Madison, USA) according to the manufacturer's instructions. In brief, primary hepatocytes isolated from mice fed a normal chow diet or a HFD were seeded in 96-well plates at a density of $5 \times 10^4$ cells per well and harvested in phenol red-free medium after culture for 48 h. Cell culture media were then collected and used to measure the

amount of LDH. The level of LDH was determined by measuring the absorbance at 490 nm with a microplate reader. For the apoptosis assay, cells were harvested and collected by centrifugation. Cells were washed with PBS and resuspended in 500 μL of binding buffer. After 5 μL each of Annexin V-FITC and propidium iodide (Beyotime, Nantong, China) were added, the cells were incubated in the dark for 10 min and then subjected to flow cytometric analysis.

**Isolation and characterisation of hepatic EVs**. Bovine EV-depleted medium was obtained by overnight ultracentrifugation at $100,000 \times g$ of medium supplemented with 20% FBS or CS. Primary hepatocytes were washed in PBS (Gibco) and further cultured in EV-depleted medium (10% FCS) for 48 h before the conditioned medium was collected for EV isolation. The media were recovered and centrifuged at $600 \times g$ at 4 °C for 5 min to remove whole cells. The cell debris and apoptotic bodies in the medium were removed by centrifugation at $1200 \times g$ at 4 °C for 15 min. The resulting medium was centrifuged at $10,000 \times g$ at 4 °C for 30 min and then filtered through a 0.2-μm filter to remove large EVs. Clarified samples were then concentrated in 100 KD MWCO Amicon Ultra-15 centrifugal filter units and then subjected to ultracentrifugation at $100,000 \times g$ for 1.5 h at 4 °C to pellet EVs. After the pellet was washed with PBS ($100,000 \times g$ for 20 min), it was resuspended in PBS, and EVs were characterised by measuring the expression of the EV-associated protein markers CD63, CD81, TSG101 and Syntenin-1 with western blotting. The EV concentration and size distribution were determined by dynamic light scattering technology using a Particle Metrix ZetaView Nanoparticle Tracking Analyser (detection range from 50 nm to 1000 nm diameters). To visualise EVs by transmission electron microscopy (TEM), 5 μL of the EV sample was fixed over-night in 4% paraformaldehyde, and 5 μL of the fixed EV solution was transferred to Formvar and carbon-coated copper grids. The grids were covered and incubated for 10 min before they were washed once with distilled water and stained with 2% uranyl acetate prior to visualisation[49]. For all EV-related experiments, EV-free FBS or CS was used.

**Sucrose gradient fractionation analysis of EVs**. Isolated EVs were analysed by sucrose gradient centrifugation according to a previously published protocol[16]. Crude EV preparations were diluted in 82% sucrose and layered at a sucrose gradient ranging from 82–10% (1.002–1.34 g per ml calculated density) as follows: 10–16% (F1), 22–28% (F2), 34–40% (F3), 46–52% (F4), 58–64% (F5) and 70–82% (F6). Samples were centrifuged at $100,000 \times g$ for 8 h. The fractions were collected, diluted 1:100 in PBS and centrifuged at $100,000 \times g$ for 1.5 h to pellet any EVs for western blot analysis. Samples were loaded into sodium dodecyl sulphate-polyacrylamide gel electrophoresis gels by volume normalisation.

**In vivo and in vitro treatment with EVs**. For in vitro treatment, 5 μg per mL EVs, as determined by protein measurement, were added to $1 \times 10^5$ recipient cells. To monitor EV trafficking in vitro, EVs were labelled with PKH67 fluorescent dye using a PKH67 fluorescent cell linker kit (Sigma-Aldrich, USA). After PKH67 staining, the EVs were purified to wash out PKH67 micelles by density gradient centrifugation[50]. Finally, PKH67-labelled EVs were resuspended in PBS. 3T3-L1 preadipocytes or L6 myocytes were incubated with PKH67-labelled EVs for 5 h. To monitor the trafficking of EVs containing miRNAs in vitro, EVs secreted by WT hepatocytes transfected with Cy3-labelled let-7e-5p were labelled with a PKH67 fluorescent dye and incubated with 3T3-L1 cells. Nuclei were visualised by staining with Hoechst 33258 (MCE, USA), and the cells were visualised with an Operetta High Content Imaging system (PerkinElmer, USA). A previous study showed that the quantity of circulating EVs in obese mice is ~ 30 μg per mouse[26]. Therefore, for in vivo treatment, 30 μg of hepatocyte-derived EVs were adoptively transferred into recipient mice via tail vein injection or iWAT injection.

**Metabolic parameter measurement**. For the glucose tolerance test, mice were injected intraperitoneally with D-glucose (2 mg per g body weight) after an overnight fast, and tail blood glucose levels were monitored every 0.5 h using a glucometer monitor (Roche). For the ITT, mice were injected intraperitoneally with human insulin (Eli Lilly) (0.75 mU per g body weight) after a 4 h fast, and tail blood glucose was monitored every 0.5 h.

**Metabolic measurement**. The oxygen consumption rate was measured with a Seahorse XF24 extracellular flux analyser according to the manufacturer's instructions.

**EdU click assay**. After 6d-differentiated 3T3-L1 preadipocytes were challenged with EVs from mice fed a HFD for 6 h or 12 weeks, the cells were stained with EdU. The number of EdU-positive cells was measured at 488 nm according to the instructions of the EdU Click Kit (Sigma-Aldrich, USA).

**Lipogenesis assay and lipolysis assay**. For the lipogenesis assay, mature adipocytes from iWAT and 3T3-L1 adipocytes were isolated to measure lipid accumulation by a colorimetric assay (Cayman Chemical Company, MI, USA)[51]. Approximately 75 μL of Lipid Droplets Assay Fixative (Cayman Chemical Company, MI, USA) was added to each well and incubated for 15 min. Wells were washed with a wash solution twice for 5 min each and left to dry completely by placing the plate under a blowing hood. Dye extraction solution was added, the wells were gently mixed for 20 min and the degree of lipogenesis was quantified from lipid droplets in cells by obtaining the absorbance at a single fixed wavelength of 490 nm with a microplate reader. For the lipolysis assay, isolated adipocytes from iWAT were incubated in phenol red-free DMEM supplemented with 2% fatty acid-free bovine serum albumin (BSA) for the indicated time at 37 °C in the presence or absence of 10 μM isoproterenol. The glycerol content in the medium was measured colorimetrically at 540 nm according to the instructions of the Lipolysis Colorimetric Assay Kit (Sigma-Aldrich, USA) against a set of glycerol standards. The cells were then washed with ice-cold PBS and lysed in 1% Triton X-100 buffer, and the protein concentration was determined and used to normalise glycerol release. All experiments were carried out in triplicate.

**Fluorescent imaging**. For in vivo fluorescent imaging, EVs were labelled with DiI (1,1′-Dioctadecyl-3,3,3′,3′-tetramethylindocarbocyanine perchlorate; Sigma-Aldrich, USA). DiI was added to an EV-PBS mixture at a final concentration of 1 μM and was incubated for 20 min before spin washing, followed by an additional wash to remove excess dye. Red-labelled EVs were injected via tail vein into WT or LKO mice fed a normal chow diet or a HFD, and whole-body red fluorescence was acquired on an IVIS Spectrum system (Caliper Life Sciences, USA). In the control groups, empty liposomes (FormuMax) were used[22]. The mice were killed 6 h later, their organs were prepared, and the red fluorescence was quantified in the liver and adipose tissue. The fluorescence intensity was measured using Living Image 3.1 software (Caliper Life Sciences, USA).

**MiRNA expression analysis**. For the microarray analysis, the miRNA components in primary hepatocyte-derived EVs from WT and LKO mice fed a normal chow diet or a HFD were profiled. Each sample comprised a pool of EVs derived from the primary hepatocytes of four animals. Total RNA from each pooled sample was isolated using mirVanaTM PARISTM (Cat #AM1556, Ambion, Austin, TX, USA) for Agilent miRNA microarray analysis (Shanghai Biotechnology Corporation). Procedures were performed as described on the Shanghai Biotechnology Corporation website (http://www.ebioservice.com). Raw data were normalised to U6 levels and analysed using GenePix Pro 4.0 software (Axon Instruments, USA). The following criteria were used to screen the miRNAs from the array data set: miRNAs with a signal intensity > 30 were selected to avoid weak signal data; miRNAs from the HFD-EVs or Chow-LKO EVs groups were each compared with Chow-EVs group; and miRNAs that showed opposite expression ratios in the HFD-EV and Chow-LKO EV groups after normalisation. The data are presented as a heat map with a colour indicating the fold change for each miRNA. Quantitative

real-time PCR analysis was used to verify miRNA expression. The cellular or EV-based miRNA expression was measured[52]. Data were normalised to levels of U6 (cellular), cel-miR-39-3p (plasma) or total protein from EVs as appropriate. All reagents for stem-loop RT were obtained from RiboBio (China). Primers and other reagents for mature miRNA assays were purchased from RiboBio (China).

**Adenoviral infection and MicroRNA or SiRNA transfection**. 3T3-L1 pre-adipocytes were incubated with adenovirus expressing either sh-Pgc1α or scrambled shRNA overnight in growth medium. The medium was then replaced, and cells were maintained in growth medium for an additional 36 h before miRNA transfection. Transfections were performed at a concentration of 20 nM for the mimics or antagomirs. After 4 h, the medium treating the transfection complex was replaced with fresh adipogenic induction medium. After 2 days of induction, the medium was replaced with adipogenic maintenance medium, and the cells were collected for RNA analysis after an additional 4 days of differentiation. All experiments were performed in triplicate for each condition and independently repeated four times. Mimics, antagomirs and siRNA were transfected into recipient cells with Lipofectamine RNAiMAX reagent (ThermoFisher Scientific, USA). After 24 h, the transfection efficiencies were validated by either qPCR or western blot analysis. The miR-mimics and antagomirs were purchased from RiboBio (China).

**Luciferase assay**. Plasmids carrying the Renilla luciferase gene linked to a fragment of the Pgc1α 3′-UTR harbouring let-7e-5p putative-sites were co-transfected into 3T3-L1 cells along with either control miRNA or let-7e-5p mimic (RiboBio, China). A mutant 3′-UTR of Pgc1α was constructed by site-directed mutagenesis of let-7e-5p from AUGGAG into GUAACG. 3T3-L1 cells were cultured in DMEM (Gibco, USA) containing 10% CS and seeded in 12-well plates. At 24 h after plating, 0.2 mg of firefly luciferase reporter plasmid, 0.2 mg of pGL3-basic expression vector (Promega, USA) and equal amounts (20 pmol) of let-7e-5p mimic or scrambled negative control RNA were transfected into cells with Lipofectamine 2000 (Invitrogen, USA) according to the manufacturer's instructions. A pGL3-basic vector was used as a transfection control. At 24 h post transfection, the cells were analysed using a luciferase assay kit (Promega, USA). All experiments were performed in triplicate wells for each condition and repeated three times independently.

**Immunofluorescence microscopy**. Cells ($1 \times 10^5$) were seeded on glass coverslips for 18 h, fixed in 4% paraformaldehyde, quenched with 0.1 M glycine, permeabilized in 0.2% BSA-0.05% saponin in PBS and incubated with primary antibodies targeting CD63 (BD Biosciences, USA) followed by treatment with fluorescent-labelled secondary antibody and DAPI (Santa Cruz Biotechnology, USA).

**Subcellular fractionation**. Subcellular primary hepatocyte fractionation was performed using the Triton X-114 partition method and ultracentrifugation[29]. In brief, hepatocytes with the indicated treatment were lysed in 500 μl lysis buffer. The total protein concentration was diluted to 1 mg/ml and partitioned with same volume of 4% Triton X-114 for 5 min at 37 °C to solubilise and fractionate the lipid-rich cell membrane. The aqueous upper phase contains enriched intracellular protein, and the organic lower phase contains highly enriched membrane-associated proteins. All the samples were subjected to immunoprecipitation and western blot detection of the proteins that were present at different portions.

**Co-immunoprecipitation analysis of Ago-RNA complex**. First, let-7e-5p$^{-/-}$ adipocytes were treated with EVs from hepatocytes transfected with mimic-NC or let-7e-5p mimic and then lysed in complete lysis buffer. Then, the total protein equivalent to $1.5 \times 10^7$ cells was incubated with pan-Ago antibody at 4 °C overnight in immunoprecipitation (IP) buffer for Proteinase K digestion and RNA purification. Negative control mouse IgG was used as an IP control. Ago protein immunoprecipitation for western blot analysis and purification of total RNA associated with Ago were performed with a miRNA Target IP Kit (Active Motif) according to the manufacturer's instructions.

**Western blot analysis**. Cells or tissues were homogenised in radio-immunoprecipitation assay buffer supplemented with protease and phosphatase inhibitors (ThermoFisher Scientific, USA). Equal amounts of protein lysate from each biological replicate were subjected to western blotting. Primary antibodies targeting CD63 (Abcam, 1:1000; ab68418), TSG101 (Abcam, 1:1000; ab125011), Syntenin-1 (Abcam, 1:1000; ab205861), Rab5 (Abcam, 1:1000; ab18211), Rab27A (Abcam, 1:1000; ab55667), CD81(Santa Cruz Biotechnology, 1:1000; sc-166029), Pgc1α (Santa Cruz Biotechnology, 1:1000; sc-518038), GGPPS (Santa Cruz Biotechnology, 1:200; sc-271680), β-actin (Santa Cruz Biotechnology, 1:1000; sc-47778), Calnexin (Cell Signaling Technology, 1:1000; 2433), CALR (Cell Signaling Technology, 1:1000; 12238), H3 (Cell Signaling Technology, 1:1000; 4499) and AGO2 (Proteintech, 1:500; 10686-1-AP) were used. Western blot data in the figures and supplemental figures are all representative of more than three independent experiments. The uncropped and unprocessed scans of the blots were provided in the Source Data file.

**RNA extraction, reverse transcription and qRT-PCR analysis**. RNA from tissues was isolated using TRIzol reagent (Takara Bio, Japan) and reverse transcribed using PrimeScript RT Master Mix for RT-PCR (TaKaRa Bio, Japan). Real-time PCR was performed using SYBR Select Master Mix (Applied Biosystems, USA) on an ABI 7300 sequence detector (Applied Biosystems, USA). The results were normalised to the 18S level in each sample, and endogenous U6 small nuclear RNA or cel-miR-39-3p were used for normalisation of cellular or plasma miRNA. All reactions were carried out in triplicate, and analysis was carried out using the $2^{-\Delta\Delta Ct}$ method. A complete list of PCR primers is shown in Supplementary Table 2.

**Quantification and statistic analysis**. The results are expressed as the mean ± SEM or SD. All cell culture and qPCR data points presented here are the average of technical duplicates. For primary cell culture experiments, the $n$ value corresponds to a cell preparation from separate mice. For adipocyte and myocyte cell culture experiments, the $n$ value corresponds to an experiment conducted with separate dishes/well numbers. For all mouse studies, the $n$ value corresponds to individual mice of a given treatment. The statistical significance of the differences between various treatments was measured by either the unpaired Student's $t$ test or one-way analysis of variance with Bonferroni post test. Data analyses were performed using GraphPad Prism software version 6.0. $P$ values from 0.001 to 0.05 or 0.001 were considered significant (*) or very significant (**/## or ***/###), respectively. 'NS' indicates no significance. In all the experiments reported in this study, no data points were excluded. All data points are represented in the figures and were used in the statistical analyses. There was no blinding, and no particular randomisation method was used to assign individuals to the experimental groups. Statistical analysis was performed using groups with similar variance. Limited variance was observed within sample groups.

**Reporting summary**. Further information on research design is available in the Nature Research Reporting Summary linked to this article.

## Data availability

All miRNA microarray data that support the findings of this research have been deposited in the Gene Expression Omnibus (GEO) and are accessible through the GEO accession number GSE141091 (https://www.ncbi.nlm.nih.gov/geo/query/acc.cgi?acc=GSE141091). The source data underlying Figs. 1a–c, 2c–d, 3a, g, i, 4b, c, d, g, 5c, f and 6j, k and Supplementary Figs 1c, g–i, 3b, c, n, 4b, c, e–h, 5e, 6c, 7f, k and 8a are provided as a Source Data file. All other relevant data of this study are available from the corresponding authors upon reasonable request. A reporting summary is available as a Supplementary Information file.

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

## Acknowledgements

This work was supported by the National Natural Science Foundation of China (31601153 to Y.Z., 31771572 to B.X., 91857109 to C.L.), the Natural Science Foundation of Jiangsu Province (BK20160619 to Y.Z.), the Social Development Fund of Jiangsu Province (BE2017708 to C.L.), the China Postdoctoral Science Foundation (2016M601778 to Y.Z.) and the Fundamental Research Funds for the Central Universities (14380269, 14380343 to Y.Z.). We thank Translational Medicine Core Facilities, Medical School of Nanjing University and Department of Hepatobiliary Surgery, Nanjing Drum Tower Hospital, The Affiliated Hospital of Nanjing University Medical School for kindly providing service and human samples.

## Author contributions

Y.Z., B.X. and C.L. developed the study concept and experimental design; Y.Z., M.Z., S.J., J.W., J.L., X.Y., D.S., J.Z., N.Z. performed experiments; X.S., L.F. and J.H. collected and helped analyse the human samples. Y.Z. and C.L. interpreted the data and wrote the manuscript.

## Competing interests

The authors declare no competing interests.
