## [Peer Review File · Nature Communications]

Editorial Note: Parts of this Peer Review File have been redacted as indicated to remove references to third-party material for which we do not have permission to publish.

Reviewers' Comments:

Reviewer #1:

Remarks to the Author:

Zhao et. al. report that hepatocytes undergoing acute lipid overload by high fat feeding release exosomes, which are taken up by adipocytes and promote adipocyte lipid storage. Hepatocyte exosome release is regulated by the Ggpps protein. miRNAs in the hepatocyte-derived exosomes are responsible for changes in adipose tissue remodeling. Exosomal let-7-5p induces adipocyte lipid storage to a greater degree than other miRNAs identified. This is accomplished by a let-7-5p-dependent increase in lipogenesis and a decrease in mitochondrial biogenesis.

This manuscript is combines in vivo and in vitro techniques to convincingly demonstrate the role of hepatocyte exosomes in adipocyte function. The experimental design is thorough and provides important new insights into how the liver and adipose tissue coordinate the appropriate storage of dietary lipids. The following concerns should be addressed to provide clarity and solidify the conclusions made:

1) Under all conditions where adipocytes in culture or adipose in vivo are being treated with hepatocyte-derived exosomes, the authors demonstrate increased adipocyte lipogenic genes and, in some instances, they also demonstrate an increase in adipogenesis-regulated genes, like PPARgamma and adiponectin. It seems possible that the hepatocyte exosomes are enhancing adipogenesis not just lipogenesis. In fact, increased lipogenesis could just be a consequence of increased adipogenesis. This is particularly important in the 3T3L1 adipocyte experiments were, in some cases, preadipocytes were treated with exosomes following just 2d of differentiation. Oil red O was used to demonstrate increased lipid droplet formation, which could also be argued as evidence for adipogenesis. The authors mention that let-7e-5p is known to promote adipogenesis (page 23 line1-2). Do the authors see a change in adipocyte number in the in-vivo experiments? The authors could do a more rigorous exploration of adipogenic gene expression in adipocytes or adipose tissue treated with hepatocyte exosomes. As for the 3T3L1 cells, the authors should wait for the cells to become fully differentiated and treat them with exosomes during the maturation stage where the lipid droplets are increasing in size. If exosomes no longer stimulate further lipid deposition, the effects of exosomes may be related to enhancing adipogenesis, not just lipogenesis. Demonstrating these exosome-stimulated adipogenesis events would give this story greater impact, as adipose tissue expansion by adipogenesis leads to a much healthier metabolic phenotype than expansion by hypertrophy. The field is also in search of mechanisms to "unlock" the differentiation potential of adipose tissue to promote healthier systemic metabolism.

2) Similarly, is not clear if the authors are promoting this mechanism as "healthy" or "unhealthy". They demonstrate this occurs in both acute lipid overload and more chronic HFD or in human obesity. An easy way to address this may be to do an oral glucose tolerance test in wt and liver Ggpps KO mice to determine if not having this mechanism is beneficial or detrimental. At the least, the authors should discuss this topic.

3) The authors claim that a major part of the mechanism is the ability of let-7e-5p to reduce PGC1a. Firstly semantic issues: Pg2. Line 14 "directly" should be indirectly because enhanced fatty acid oxidation will indirectly effect adipocyte lipid accumulation. Additionally, the authors should avoid claims that the effect of let-7e-5p is reducing fatty acid oxidation because they did not provide any experimental evidence for that. The authors could run an oxygen consumption experiment on adipocytes treated with hepatocyte exosomes or demonstrate that exosomes actually decrease adipocyte mitochondria number (western blot for mitochondrial proteins or quantification of mitochondrial DNA). Lastly, the overall result of the exosomes to increase fat pad weight is not just

due to reduced PGC1a expression, but also increased lipogenesis. That should be made clear in the text.

- 4) The author's interpretation of the role of Ggpps in exosome secretion is a stretch. i) Fig 4D. in the rab27a mutant-expressing cells it looks like CD63 is being miss-localized to the plasma membrane, not necessarily that there is reduced number of CD63 positive vesicles. The authors should use more sensitive imaging techniques if they want to demonstrate a reduction in endosome formation. Similarly, the claim that Ggpps siRNA reduces exosome production as measured by immunofluorescence is not substantiated (Figure 3B). This claim cannot be correct unless quantification of exosomes in the media is provided, as exosomes are, by definition, extracellular. Immunofluorescence can only tell us that there is less expression of CD63 or less endosome formation. Regardless of these technical issues, it seems clear that Ggpps does promote exosome secretion, however, it looks like it is doing this by altering endosomal function. This means there may be many other vesicle trafficking-related consequences of a knock out for Ggpps in the hepatocyte, not just exosome secretion. For example, is exosome secretion reduced simply because the hepatocyte can't function properly? That needs to be further investigated.
- 5) Figure 4E and S4E: The liver from KO mice displayed increased uptake of exogenously injected hepatocyte exosomes, whereas adipose tissue takes up less. The purpose of this experiment is not clear. Are the authors trying to connect liver Ggpps to homing of exosomes? It seems out of place and the rationale or significance of the result in testing the hypothesis is not mentioned.
- 6) Other minor suggestions:
- The authors should avoid using vague statements that are not entirely true. Example: Pg2 line 7, Pg3 line 8-9- the claim that the mechanisms that allow the liver to communicate with other organs are unknown is not true. We have many examples of crosstalk between liver and other organs, particularly adipose tissue. The novelty here is this is not mediated by a soluble hepatokine, but is exosome driven.
 - Fig S2 page 9 line 19-20: sentence is confusing. It is not clear that the authors isolated exosomes from serum. It can be interpreted to mean they isolated exosomes from conditioned media.
 - Fig 4B: the authors use the right markers for showing the presence of exosomal particles, however, it is just as important to show the absence or lower abundance of other cellular proteins. The blot for calnexin should be expanded to show the calnexin expression in the exosomes compared to the parental cells. We should see exosomes are enriched with CD63 and TSG101 but contain less for calnexin.
 - Page 22 line 10 Should it be "hepatocytes from HFD-fed mice" instead of "hepatocytes under HFD"?

Reviewer #2:

Remarks to the Author:

I have several major concerns.

Exosomes are vesicles secreted when membranes of multivesicular endosomes fuse with the plasma membrane. Because there are no physical (e.g. diameter, density) or biochemical (e.g. presence of tetraspanins) features anymore that distinguish exosomes from other types of vesicles, I recommend to use the common umbrella term "extracellular vesicles" (EVs). The authors do not provide any evidence that they study exosomes or only exosomes.

Many data are shown as "Relative" (for example Fig 2A, 2B, 2I, etc.), but it is unclear what the absolute values are, and whether such values differ per condition. This makes the interpretation of the results difficult if not impossible. Furthermore, many controls are lacking. Reliably measuring small particles as EVs has proven exceptionally difficult, and controls are essential. For example, when measuring concentrations or diameter of EVs isolated from CM, also Chow or HFD media themselves should be taken alongside as controls, including ultracentrifugation procedure etc., to exclude that observed EVs originate from the CM (containing Chow or HDM) itself.

The authors isolate EVs by ultracentrifugation and find the EV-containing fraction “enriched” for miR-122. I have two problems here. Firstly, for me the term “enrichment” means a higher concentration of –in this case– miR-122 in a given volume. Because HFD treatment of hepatocytes produces more and also larger (in diameter) EVs (Fig 2E), this can be questioned and seems not true. There simply seem to be more and larger EVs. Secondly, for example in a body fluid as human plasma only 1% or so of all “circulating miRs” are associated with EVs (PNAS, 2014). Because ultracentrifugation will also pellet non-vesicular particles and proteins/protein aggregates, and because miRs also associate with non-vesicular particles and proteins, contamination of potential non-EV carriers of miR-122 should be excluded.

The authors incubated the primary hepatocytes with Chow or HFD. I think it is essential to exclude both induction of cell lysis (e.g. by measuring release of LDH) as well as induction of apoptosis (which may also explain the shift in diameter of EVs as observed in Fig 2E).

Apart from the fact that the resolution of the TEM image is too low to judge (Fig 2C), the picture shown is quite interesting. To me, most particles shown look very angular, which is not typical for EVs. When we suppose the small events shown are indeed EVs, their diameter seems to be mostly between 30-50 nm, which clearly differs from the data shown in Fig 2E (mean diameter 80-90 nm). The picture also clearly shows some big white blobs, which to me look like fat particles (chylomicrons perhaps?). This confirms that one should be careful by overinterpretation (“highly enriched”, as claimed by the authors on page 9) of the presence of miR-22 in EVs (see previous remark about enrichment and ultracentrifugation).

The lower limit of detection of the Malvern equipment should be taken into account, because it seems unlikely to me that single particles with a low refractive index of 30-50 nm would be detectable.

On page 38, the methodology of “exosome purification and characterization” is described. A few brief questions. Most if not all culture media contain a physiological concentration of calcium ions. Calcium together with phosphate (in the used PBS) is insoluble (calcium phosphate crystals being formed). Why was PBS chosen? Regarding the isolation of CD63-exposing EVs, to the best of my knowledge nobody working in the EV field has any positive experience so far with capturing beads for EVs. Capturing leads to subselection, and isolated EVs are a tiny fraction usually of all EVs present. Thus, the “EVs” shown in this TEM image are likely not representative for the (total) population of EVs as shown in the other experiments throughout this manuscript. Thus, it is essential to show representative TEM pictures of (all and thus not only CD63-selected) EVs present in the CM, because otherwise data are incomparable between experiments, and make interpretation impossible.

For in vitro treatment, exosomes are quantified based on measuring the concentration of protein. However, most of the measured protein –due to the crude concentration method applied (ultracentrifugation)– is likely to be from contaminants rather than from the EVs themselves.

The described procedure for staining with PKH67 seems wrong, because PKH67 is well known to form micelles, which need to be separated from the EVs after labelling by density gradient centrifugation (a detailed protocol has been published by the group of Marca Wauben in Nature Protocols a few years ago). To which extent this also holds true for DiI I do not know, but recent papers (see e.g. de Rond, Clin Chem 2018) do not give much hope because all these so called “general dyes” stain not only EVs but also lipoproteins and even proteins. Without appropriate controls, these data cannot be correctly interpreted.

Reviewer #3:

This study hypothesizes that the liver secretes miRNAs in exosomes and communicates to the AT regulating adipocyte metabolism. The authors demonstrate that the geranylgeranylation of hepatic proteins influences exosome biogenesis and avoid the communication between the liver and AT via exosomes. While the concept is potentially interesting, there are key experiments to support this model including:

- 1) Demonstration of the direct transfer between the liver and AT by measuring the expression of liver specific miRNAs (miR-122) in the adipose tissue.
- 2) Analysis that the transfer of miRNAs regulates adipocyte gene expression in vivo. Then authors should pull down the Ago complex and assess the relative enrichment of hepatic derived miRNAs and the mRNAs target within the RISC complex.
- 3) The metabolic analysis in the Ggpps is very descriptive. The authors should perform a deep analysis of how absence of hepatic Ggpps influence metabolic parameters (metabolic cages study) and whole glucose metabolism (hyperinsulinemic/euglycemic clamps)
- 4) RNA-Seq analysis of AT isolated from mice fed a HFD.
- 5) Overall, the authors should perform key experiments under physiological and in vivo conditions to define the relevance of this study.

Reviewer Comments:

Reviewer #1:

REVIEWER1, COMMENT: Zhao et. al. report that hepatocytes undergoing acute lipid overload by high fat feeding release exosomes, which are taken up by adipocytes and promote adipocyte lipid storage. Hepatocyte exosome release is regulated by the Ggpps protein. miRNAs in the hepatocyte-derived exosomes are responsible for changes in adipose tissue remodeling. Exosomal let-7-5p induces adipocyte lipid storage to a greater degree than other miRNAs identified. This is accomplished by a let-7-5p-dependent increase in lipogenesis and a decrease in mitochondrial biogenesis.

This manuscript is combines in vivo and in vitro techniques to convincingly demonstrate the role of hepatocyte exosomes in adipocyte function. The experimental design is thorough and provides important new insights into how the liver and adipose tissue coordinate the appropriate storage of dietary lipids.

RESPONSE: We were greatly encouraged by the reviewers' largely positive comments of our work. We appreciate the constructive comments of the reviewer that the different effect of hepatic EVs from acute and chronic lipid overload on adipose hyperplasia would largely enforce our observation. Based on these suggestions, we have performed several experiments and address the concerns in a point-by-point response below. More importantly, our findings about hepatic sEVs-dependent adipogenesis in adipose will "unlock" the differentiation potential of adipose tissue and give our work greater impact. THANKS!

REVIEWER1, COMMENT 1: The following concerns should be addressed to provide clarity and solidify the conclusions made:

Under all conditions where adipocytes in culture or adipose in vivo are being treated with hepatocyte-derived exosomes, the authors demonstrate increased adipocyte lipogenic genes and, in some instances, they also demonstrate an increase in adipogenesis-regulated genes, like PPARgamma and adiponectin. It seems possible that the hepatocyte exosomes are enhancing adipogenesis not just lipogenesis. In fact, increased lipogenesis could just be a consequence of increased adipogenesis. This is particularly important in the 3T3L1 adipocyte experiments were, in some cases, preadipocytes were treated with exosomes following just 2d of differentiation. Oil red O was used to demonstrate increased lipid droplet formation, which could also be argued as evidence for adipogenesis. The authors mention that let-7e-5p is known to promote adipogenesis (page 23 line1-2). Do the authors see a change in adipocyte number in the in-vivo experiments? The authors could do a more rigorous exploration of adipogenic gene expression in adipocytes or adipose tissue treated with hepatocyte exosomes. As for the 3T3L1 cells, the authors should wait for the cells to become fully differentiated and treat them with exosomes during the maturation stage where the lipid droplets are increasing in size. If exosomes no longer stimulate further lipid deposition, the effects of exosomes may be related to enhancing adipogenesis, not just lipogenesis. Demonstrating these exosome-stimulated adipogenesis events would give this story greater impact, as adipose tissue expansion by adipogenesis leads to a much healthier metabolic phenotype than expansion by hypertrophy. The field is also in search of mechanisms to "unlock" the differentiation potential of adipose tissue to promote healthier systemic metabolism.

RESPONSE: We appreciate the reviewer's efforts to demonstrate the sEVs-stimulated adipogenesis event. We agree with the reviewer that it will give this work greater impact. Based on the recommendations of the reviewer, we checked the adipocyte number and adipogenesis gene expression *in vivo* after dietary lipid challenge and found that the adipogenesis genes in iWAT increased by 24 hours after HFD treatment, while adipocyte number increased by 1 week (Response Fig1). We then challenged fully differentiated 3T3-L1 preadipocytes (differentiated for 8 day, the stage that adipogenesis has finished but lipogenesis is proceeding) with EVs from 6-hour-HFD treated mice for 2 days. The results showed that there was no significant change of both TG accumulation and lipogenesis gene expression (Response Fig2). Our data indicate that, as the reviewer1 suggested, the short-term lipid overload drove liver to induce adipose hyperplasia by adipogenesis through releasing hepatic EVs, leading to a much

healthier systemic metabolism to relieve the pressure by acute lipid overload. Thus, the hepatic sEVs-stimulated adipogenesis by the short-term lipid overload may “unlock” the differentiation potential of adipose tissue to promote healthier systemic metabolism.

Response Fig1 (Revised Fig1J and 1K). The adipocyte number and adipogenesis genes expression in iWAT and eWAT after lipid overload. (J) Quantification of adipocyte number of iWAT and eWAT from HFD-fed mice at different time points. (K) Expression of fatty acid transport, lipogenesis and fatty acid oxidation-related genes in the AT of mice following HFD exposure or indicated time.

Response Fig2 (Revised Fig2I-J). The effect of HFD-6h sEVs on adipocyte. (I) Hepatocyte-derived sEVs from mice treated with 6-h Chow diet/HFD were added to 3T3-L1 preadipocytes differentiating for 8 days. Adipocytes differentiated from 3T3-L1 adipocytes at day 10 after induction were stained with Oil Red O. (J) Expression of adipogenesis and lipogenesis genes in 3T3-L1 preadipocytes from Fig2I.

REVIEWER1, COMMENT 2: Similarly, it is not clear if the authors are promoting this mechanism as “healthy” or “unhealthy”. They demonstrate this occurs in both acute lipid overload and more chronic HFD or in human obesity. An easy way to address this may be to do an oral glucose tolerance test in wt and liver Ggpps KO mice to determine if not having this mechanism is beneficial or detrimental. At the least, the authors should discuss this topic.

RESPONSE: We thank the reviewer for pointing out the need to analyze the metabolic status in WT and liver *Ggpps* KO mice to determine the effect of hepatic-sEVs by chronic HFD. By

glucose tolerance test and insulin tolerance test, we do find that hepatic *Ggpps* deletion could improve systemic glucose tolerance and insulin sensitivity (Response Fig3), suggesting that blocking the hepatic-sEVs secretion is beneficial. We have discussed this in Page 41, Line 6-8 “blocking *Ggpps*-dependent sEVs secretion upon chronic lipid overload was beneficial by reducing adipocyte mass (Fig3) and improving systemic glucose tolerance and insulin sensitivity.”.

Response Fig3 (Revised Fig4E-F). The glucose tolerance tests and insulin tolerance tests in LKO Mice. Glucose tolerance tests (GTT) (E) and insulin tolerance tests (ITT) (F) were performed in WT and LKO mice.

REVIEWER1, COMMENT 3: The authors claim that a major part of the mechanism is the ability of *let-7e-5p* to reduce PGC1a. Firstly semantic issues: Pg2. Line 14 “directly” should be indirectly because enhanced fatty acid oxidation will indirectly effect adipocyte lipid accumulation. Additionally, the authors should avoid claims that the effect of *let-7e-5p* is reducing fatty acid oxidation because they did not provide any experimental evidence for that. The authors could run an oxygen consumption experiment on adipocytes treated with hepatocyte exosomes or demonstrate that exosomes actually decrease adipocyte mitochondria number (western blot for mitochondrial proteins or quantification of mitochondrial DNA). Lastly, the overall result of the exosomes to increase fat pad weight is not just due to reduced PGC1a expression, but also increased lipogenesis. That should be made clear in the text.

RESPONSE: We regret the lack of a clear description in the initial submission. We correct” the imprecise description “directly” to “increasing lipogenesis and inhibiting lipid oxidation” (Revised Page2, Line16). We retained the phrase “claims that the effect of *let-7e-5p* is reducing fatty acid oxidation” since we have now provided direct experimental evidence to confirm *let-7e-5p* reduced fatty acid oxidation. Firstly, an oxygen consumption experiment on adipocytes indicated that the mitochondrial respiration of adipocytes by utilization of fatty acid was substantially inhibited by sEV-*let-7e-5p*-mimic. Also, adipocyte mitochondria number was decreased by sEV-*let-7e-5p*-mimic (Response Fig4). All these data show that sEVs containing *let-7e-5p* increase lipogenesis and reduce lipid oxidation.

Lastly, as we have found in response figure 1, the short-term lipid overload could enhance adipogenesis through hepatic EVs which contains both miRNA-210-3p and let-7e-5p. It has been reported that miRNA-210-3p could regulate adipocyte differentiation through Wnt signaling (Qin, L. *et al BMC genomics*, 2010), and we have found let-7e-5p could reduce fatty acid oxidation. Thus, sEVs contained miRNA210-3p and let-7e-5p function together to increase adipogenesis and inhibit lipid oxidation resulting in lipid deposition in fat tissue (highlighted by red in revised Abstract Page 2, Line 16 and Fig. 8 figure legend Page 38 Line 7-8).

Response Fig4 (Revised Fig57F-G). The fatty acids oxidation and mtDNA of adipocytes treated by EV-let-7e-5p-mimic. (F) OCR of 3T3-L1 preadipocytes using fatty acids detected with the XF Palmitate-BSA FAO Substrate with the XF Cell Mito Stress Test. n=5 for each group. (G) Relative mtDNA of 3T3-L1 preadipocytes from Fig6H.

REVIEWER1, COMMENT 4-1: The author's interpretation of the role of Ggpps in exosome secretion is a stretch. i) Fig 4D. in the rab27a mutant-expressing cells it looks like CD63 is being miss-localized to the plasma membrane, not necessarily that there is reduced number of CD63 positive vesicles. The authors should use more sensitive imaging techniques if they want to demonstrate a reduction in endosome formation. Similarly, the claim that Ggpps siRNA reduces exosome production as measured by immunofluorescence is not substantiated (Figure 3B). This claim cannot be correct unless quantification of exosomes in the media is provided, as exosomes are, by definition, extracellular. Immunofluorescence can only tell us that there is less expression of CD63 or less endosome formation.

RESPONSE: We regret for not mentioning the purpose of immunofluorescence imaging clearly. We did this experiment to see the effect of Ggpps on distribution of MVEs, not endosome formation. Based on the suggestion of the reviewer, we re-examined the distribution of CD63-positive MVEs by immunofluorescence. Ggpps-knockdown cells and Rab27A mutant-expressing cells showed the clustered distribution of MVEs, indicating that Ggpps influence the MVE intracellular location (Response Fig5). We have discussed this in Page 16, Line 3 and Page 19, line21-Page 20, Line1. Besides, we also performed WB to detect the sEVs markers and quantified sEVs production by their protein level to measure the sEV secretion (Response Fig6).

Response Fig5 (Revised Fig3B and Fig4E). EV intercellular distribution by *Ggpps* deficiency in hepatocytes.

Fluorescence microscopy of primary hepatocytes treated with a scrambled control (Scramble) or a *Ggpps* siRNA (siGgpps). (B) Fluorescence microscopy of primary hepatocytes infected with empty vector or *Ggpps* vector and Rab27A or Rab27A mutant stained with anti-CD63 and DAPI (nucleus, blue). Scale bars, 5 μ m.

Response Fig6 (Revised Fig3C and Fig4A,4F,4G). EV production by *Ggpps* deficiency in hepatocytes. (A) Left: sEV production by WT and *Ggpps*-knockdown primary hepatocytes. Right: sEV production by primary hepatocytes in WT and liver-specific *Ggpps* knockout mice (LKO) fed with chow diet or HFD. (B) CD63, TSG101 and Calnexin protein levels in the parental cells and sEVs from WT and LKO mice fed with chow diet or HFD. (C) sEV production by primary hepatocytes infected with empty vector or *Ggpps* vector and Rab27A or Rab27A mutant. (D) TSG101 protein levels in the parental cells and sEVs from (C).

REVIEWER1, COMMENT 4-2: Regardless of these technical issues, it seems clear that *Ggpps* does promote exosome secretion, however, it looks like it is doing this by altering endosomal function. This means there may be many other vesicle trafficking-related consequences of a knock out for *Ggpps* in the hepatocyte, not just exosome secretion. For example, is exosome secretion reduced simply because the hepatocyte can't function properly? That needs to be further investigated.

RESPONSE: Geranylgeranyl diphosphate synthase (GGPPS) determines the balance of protein prenylation by catalyzing the synthesis of geranylgeranyl diphosphate (GGPP) from farnesyl diphosphate (FPP), both of which are used for the prenylation of proteins with CaaX motifs at their C-terminal (geranylgeranylation and farnesylation, respectively). In this study, we have found that geranylgeranylation of Rab27A, Rab5 were reduced by *Ggpps* deficiency in hepatocytes (Response Fig7), which may reduce their localization to membranes to exert their function in intracellular membrane trafficking pathway including EV secretion. Besides, *Ggpps* deletion impairs mitophagy by decreasing geranylgeranylation of Rab7 (Chaojun Li, J Pathol, 2018), which also involved in endocytic trafficking highway. Therefore, we agree with the reviewer that *Ggpps* knock out may cause other vesicle trafficking-related consequences. However, the farnesylation of GTPase may be enhanced by *Ggpps* deficiency that will compensate the loss of protein geranylgeranylation and maintain cell normal function. We have found that the deletion of *Ggpps* doesn't affect the cell activity (Wang XX et al J Exp Med. 2013). We also checked cell status when we isolate primary hepatocytes before to collect exosome (Response Fig8A and B) and found that *Ggpps* deficiency was not sufficient to affect hepatokines and cytokines secretion in primary hepatocytes (Response Fig8C, D and E). Thus, the decreased secretion of exosomes is not caused by poor cell viability.

C**D**
Response Fig7 (Revised Fig4C and 4D). Geranylgeranylation of Rab-GTPase by Ggpps in hepatocytes. (C) Membrane-associated Rab27A (lipid-soluble protein with geranylgeranylation in the detergent phase, down) and cytoplasm-associated Rab27A (water-soluble protein with ungeranylgeranylation in the aqueous phase, up) were obtained with Triton X-114 extraction and analyzed by immunoblotting. (D) Membrane-associated Rab5 (lipid-soluble protein with geranylgeranylation in the detergent phase, down) and cytoplasm-associated Rab5 (water-soluble protein with ungeranylgeranylation in the aqueous phase, up) were obtained with Triton X-114 extraction and analyzed by immunoblotting.

Response Fig8 (Revised FigS5). Cell death and cell apoptosis in WT and LKO mice. (A) LDH release by primary hepatocytes isolated from WT and LKO mice. (B) Cell apoptosis of primary hepatocytes isolated from WT and LKO mice. (C) Relative mRNA expression of circulating liver-derived hepatokines and cytokines. (D) Plasma FGF21 levels. (E) Ggpps protein expression and *Fgf21* mRNA expression in primary hepatocytes treated with *siGgpps* with or without FFA treatment.

VIEWER1, COMMENT 5: Figure 4E and S4E: The liver from KO mice displayed increased uptake of exogenously injected hepatocyte exosomes, whereas adipose tissue takes up less. The purpose of this experiment is not clear. Are the authors trying to connect liver Ggpps to homing of exosomes? It seems out of place and the rationale or significance of the result in testing the hypothesis is not mentioned.

RESPONSE: We regret this omission for mentioning the significance of the *in vivo* experiment. To verify that hepatocyte-derived EVs enter into adipose tissue *in vivo*, we injected labeled hepatocyte-derived EVs into WT and liver-specific *Ggpps* knockout mice fed with chow diet or HFD (Chow-WT, Chow-LKO, HFD-WT and HFD-LKO) through the tail vein. We found that strong hepatocyte-derived EVs signal (red fluorescence) was detected in adipose at 6h post injection, suggesting that hepatocyte-derived EVs are able to be taken up by both iWAT and eWAT, and HFD could enhance the EVs uptake by adipose. The accumulation of exogenously injected hepatocyte exosomes in liver is because of non-specific uptake of hepatocytes, which also might be affected by *Ggpps* deletion in hepatocyte. Besides, *Ggpps* deficiency in

hepatocyte resulted in less EVs taken up by adipose tissue compared with wild-type mice, suggesting that liver Ggpps might also plays critical role in liver-adipose axis (Response Fig9).

Response Fig9 (Revised Fig5B and 5C). *In vivo* fluorescence imaging by hepatocyte-derived EVs. (A) EVs of primary hepatocytes from WT mice treated with chow diet were injected into WT and LKO mice fed with chow diet or HFD through the tail vein. Empty liposomes are as controls. Red fluorescence levels in the liver, iWAT, and eWAT were acquired after 6 hours by IVIS spectrum. (C) Radiant efficiency was measured using Living Image 3.1 software in (B).

REVIEWER1, COMMENT 6: The authors should avoid using vague statements that are not entirely true. Example: Pg2 line 7, Pg3 line 8-9- the claim that the mechanisms that allow the liver to communicate with other organs are unknown is not true. We have many examples of crosstalk between liver and other organs, particularly adipose tissue. The novelty here is this is not mediated by a soluble hepatokine, but is exosome driven.

RESPONSE: We apologize for these vague statements and agree that there exist many hepatokines, like IL8, FGF21 to mediate crosstalk between liver and other organs, particularly adipose tissue. As the reviewer has correctly pointed out, we correct the initial statement to “therefore, the interplay between liver and other organs that maintains adaptive lipid homeostasis is important.” (Revised Page2 line7) as highlighted by red and delete the imprecise statement in Page3, line8-9 of initial manuscript.

REVIEWER1, COMMENT 7: Fig S2 page 9 line 19-20: sentence is confusing. It is not clear that the authors isolated exosomes from serum. It can be interpreted to mean they isolated exosomes from conditioned media.

RESPONSE: We apologize for this confusing sentence. We isolated sEVs from mice serum to detect the hepatic miR-122 expression level after acute and chronic HFD exposure. Therefore, we correct this sentence to “We also found that hepatic miR-122 (a specific miRNA enriched in hepatocytes) was increased in mice serum sEVs and adipose after acute and chronic HFD exposure (FigS3G-J).” (Revised Page11, line15-17) as highlighted by red. We hope that this is a clearer statement.

REVIEWER1, COMMENT 8: the authors use the right markers for showing the presence of exosomal particles, however, it is just as important to show the absence or lower abundance of

other cellular proteins. The blot for calnexin should be expanded to show the calnexin expression in the exosomes compared to the parental cells. We should see exosomes are enriched with CD63 and TSG101 but contain less for calnexin.

RESPONSE: We acknowledge this omission and include these data in the revised manuscript (Revised Fig4B). Based on the reviewer's suggestion, we re-examined sEV markers (CD63 and TSG101) and non-sEV protein (Calnexin) in parental cells and sEVs by Western blot (Response Fig10). The data show that sEVs contain enriched CD63 and TSG101 but less calnexin, and further validate our conclusion that *Ggpps* deficiency inhibited the sEVs secretion.

B

Response Fig10 (Revised Fig4B). The quantification of the sEVs in the conditioned medium of isolated primary hepatocytes of WT and *Ggpps* LKO mice (B) CD63, TSG101 and Calnexin protein levels in the parental cells and sEVs from WT and LKO mice fed with chow diet or HFD.

REVIEWER1, COMMENT 9: Page 22 line 10 Should it be “hepatocytes from HFD-fed mice” instead of “hepatocytes under HFD”?

RESPONSE: We regret the lack of clarity in the description. We now correct this statement to “hepatocytes from HFD-fed mice” in the revised manuscript as highlighted by red (Page23, line17).

Reviewer #2:

REVIEWER2, COMMENT 1: I have several major concerns. Exosomes are vesicles secreted when membranes of multivesicular endosomes fuse with the plasma membrane. Because there are no physical (e.g. diameter, density) or biochemical (e.g. presence of tetraspanins) features anymore that distinguish exosomes from other types of vesicles, I recommend to use the common umbrella term “extracellular vesicles” (EVs). The authors do not provide any evidence that they study exosomes or only exosomes.

RESPONSE: We agree with this assessment and we change this inaccurate statement “exosomes” to “extracellular vesicles (EVs)” in the revised manuscript as highlighted by red.

Given the overlap in size and similarity in biochemical properties, it is often difficult to experimentally distinguish exosomes from small microvesicles (Kowal et al., 2016). Thus, the general term small EV (sEV) is often used to describe this highly heterogeneous population of EVs isolated by a $100,000 \times g$ spin.

REVIEWER2, COMMENT 2: Many data are shown as “Relative” (for example Fig 2A, 2B, 2I, etc.), but it is unclear what the absolute values are, and whether such values differ per condition. This makes the interpretation of the results difficult if not impossible. Furthermore, many controls are lacking. Reliably measuring small particles as EVs has proven exceptionally difficult, and controls are essential. For example, when measuring concentrations or diameter of EVs isolated from CM, also Chow or HFD media themselves should be taken alongside as controls, including ultracentrifugation procedure etc., to exclude that observed EVs originate from the CM (containing Chow or HDM) itself.

RESPONSE: We apologize for the unclear description about the “Relative” values in the initial manuscript. I have added what the relative values refer to either in the figure legend as highlighted by red. In Fig2A, we repeated ORO staining experiment three times independently. Due to the different cell density and the time of staining of each experiment, the absolute values of absorbance are different and the variation is pretty large. Therefore, we normalized the absolute value of Chow-CM group to 1, then compared other groups to Chow-CM group and represent all data in the statistical analyses by relative values. The same considerations apply to Fig2B and 2I, in which we examined the expression of lipogenesis genes in 3T3-L1 preadipocytes treated with conditioned medium or EVs of hepatocytes three times independently. We normalized each lipogenesis gene in Chow-CM group to 1, compared the results of HFD to Chow group. The final result is shown as the fold change of each gene under HFD relative to the Chow, respectively.

According to the concern of reviewer that the primary hepatocyte’s medium should be taken alongside as controls, we repeated our experiments with exosome-free FBS or CS for all sEV-related experiments to exclude the influence of EVs from the medium itself. To exclude the interference of other extracellular vesicles, bovine EV-depleted medium was obtained by overnight ultracentrifugation at $100,000 \times g$ of medium supplemented with 20% FBS or CS. Primary hepatocytes were washed in PBS (Gibco) and further cultured in EV-depleted medium (10% EV-depleted CS final) for 48 h before collection of conditioned medium for EV isolation, which we highlight by red in the Methods (Revised Page 47-49). Therefore, we detect the pellet of sEVs-free medium itself by differential centrifugation (Revised FigS3A) and found that there is no EV abundant compared with the conditioned medium of primary hepatocyte seed for 48h (Response Fig11).

Response Fig11 (Revised FigS3D). Characteristics of sEV-free CM and CM. (D) sEV abundance in sEV-free medium itself and conditioned medium of primary hepatocytes from FigS3A.

REVIEWER2, COMMENT 3: The authors isolate EVs by ultracentrifugation and find the EV-containing fraction “enriched” for miR-122. I have two problems here. Firstly, for me the term “enrichment” means a higher concentration of –in this case- miR-122 in a given volume. Because HFD treatment of hepatocytes produces more and also larger (in diameter) EVs (Fig 2E), this can be questioned and seems not true. There simply seem to be more and larger EVs. Secondly, for example in a body fluid as human plasma only 1% or so of all “circulating miRs” are associated with EVs (PNAS, 2014). Because ultracentrifugation will also pellet non-vesicular particles and proteins/protein aggregates, and because miRs also associate with non-vesicular particles and proteins, contamination of potential non-EV carriers of miR-122 should be excluded.

RESPONSE: We apologize for the unclear description about the “enriched” in the initial manuscript. We admit that HFD treatment of hepatocytes produces more EVs based on quantification (Revised Fig2E). To make it much clearer, we have changed the “highly enrichment” to “increased” (Revised Page11 line16) as highlighted by red.

Secondly, we agree that the carriers other than EVs could mediate miRNA transfer. All EV isolation techniques potentially co-isolate other RNA-binding structures, such as large protein complexes and lipoproteins (Clotilde Théry, Cell, 2016). Therefore, as the International Society for Extracellular Vesicles highlighted (Lötvald et al., 2014), additional steps of separation of EVs from other structures, e.g., by floatation into density gradients, are necessary before claiming specific EV-mediated miRNA transfer (Clotilde Théry, Cell, 2016). Therefore, we improved the sEV isolation protocol to remove cell debris, apoptotic bodies, non-vesicular particles and proteins/protein aggregates by differential centrifugation and density gradients (Response Fig12A) as previously described (Kowal et al PNAS 2016; Jerrold M. Olefsky, Cell, 5 October 2017, Pages 372-384; Philipp E.Scherer, Cell, 18 October 2018, 175:695-708), excluding the potential non-EV carriers of miR-122 to the greatest extent (Response Fig12B).

We hope that this new isolation method will satisfy the reviewer's concern.

Response Fig12 (Revised FigS3A and S3E Right). Isolation schematic of hepatocyte-derived sEVs. (A) sWAT tissue sEV isolation schematic. (B) miR-122 levels in CM before differential centrifugation and sEVs after density gradients.

REVIEWER2, COMMENT 4: The authors incubated the primary hepatocytes with Chow or HFD. I think it is essential to exclude both induction of cell lysis (e.g. by measuring release of LDH) as well as induction of apoptosis (which may also explain the shift in diameter of EVs as observed in Fig 2E).

RESPONSE: We thank the reviewer for pointing out the need for excluding the influence of cell death and apoptosis on this hepatic sEVs-stimulated adipose remodeling event. Therefore, we measured the LDH release of hepatocytes and added Annexin V/PI to hepatocytes to

examine the cell death and apoptosis. The results show that the cell lysis and apoptosis have no significant changes between hepatocytes of mice treated with chow diet and HFD (Response Fig13).

Response Fig13 (Revised FigS2D-E). Cell death and cell apoptosis in mice after lipid overload. (D) LDH release by primary hepatocytes isolated from mice treated with chow diet and HFD by 6 hours. **(E)** Cell apoptosis of primary hepatocytes isolated from mice treated with chow diet and HFD by 6 hours.

REVIEWER2, COMMENT 5: Apart from the fact that the resolution of the TEM image is too low to judge (Fig 2C), the picture shown is quite interesting. To me, most particles shown look very angular, which is not typical for EVs. When we suppose the small events shown are indeed EVs, their diameter seems to be mostly between 30-50 nm, which clearly differs from the data shown in Fig 2E (mean diameter 80-90 nm). The picture also clearly shows some big white blobs, which to me look like fat particles (chylomicrons perhaps?). This confirms that one should be careful by overinterpretation (“highly enriched”, as claimed by the authors on page 9) of the presence of miR-122 in EVs (see previous remark about enrichment and ultracentrifugation).

RESPONSE: We regret the poor resolution of the TEM image and apologize for the overinterpretation again, so we have re-isolated sEVs by differential centrifugation and re-examined the sEVs by TEM (Revised FigS3A and Fig2C). Because we used a less rigorous method to isolate EVs in our initial manuscript, so the previous “big white blobs” may be contaminations as the reviewer mentioned in Comment 3. By electron microscopy, we now found that these EVs were morphologically heterogeneous, although the majority exhibited a cup-like structure (Response Fig14). Also, we verified these sEVs as they expressed the exosome-specific protein markers TSG101, syntenin1, CD63, largely in F2-F3 (Response Fig15B and E).

Also, we used the new isolation method to determine the size of sEVs we detected by the dynamic light scattering technology using Particle Metrix ZetaView® Nanoparticle Tracking Analyzer is around 100 nm (Response Fig15F) in accordance with previous literature (Kowal

et al PNAS 2016; Jerrold M. Olefsky, *Cell*, 5 October 2017, Pages 372-384; Philipp E.Scherer, *Cell*, 18 October 2018, 175:695-708). The sizes of hepatocyte-derived exosomes are around 100nm, which is in accordance with the data in previous findings (Seo W, et al. *Hepatology* 2016, Fig4C).

C

Primary hepatocyte-EV

Response Fig14 (Revised Fig2C). Primary hepatocyte-derived sEVs. (C)Electron microscopy analysis of sEVs secreted by primary hepatocytes from mice fed a normal chow diet. Scale bar, 200 nm.

Response Fig15 (Revised FigS3B, S3C, S3E and S3F). Characteristics of hepatocyte-Derived sEVs in mice after lipid overload. (B-C) Western blot of proteins in a sucrose gradient for sEV markers (B) or non-sEV proteins (C). Each well was loaded with 10 µg protein. (E) sEV abundance in subfractions after sucrose gradient floatation. (F) Particle characteristics of the vesicles secreted from primary hepatocytes of mice fed with chow diet or HFD were measured by using Particle Metrix ZetaView® Nanoparticle Tracking Analyzer.

REVIEWER2, COMMENT 6: The lower limit of detection of the Malvern equipment should be taken into account, because it seems unlikely to me that single particles with a low refractive index of 30-50 nm would be detectable.

RESPONSE: We acknowledge that the limit of detection of Malvern equipment may complicate some of our previous observations. To address this, we re-examined the sEVs by the dynamic light scattering technology using Particle Metrix ZetaView® Nanoparticle Tracking Analyzer, a more reliable tool which can detect the 30-50 nm particle. The NTA analysis revealed that microvesicles isolated by 100,000 × g ultracentrifugation contains abundant hepatocyte-derived EVs with a diameter of ~100 nm (Response Fig15F). [attachment redacted] provided by the company shows the single particles with a low refractive index of 30-50 nm would be detectable.

REVIEWER2, COMMENT 7: On page 38, the methodology of “exosome purification and characterization” is described. A few brief questions. Most if not all culture media contain a physiological concentration of calcium ions. Calcium together with phosphate (in the used

PBS) is insoluble (calcium phosphate crystals being formed). Why was PBS chosen? Regarding the isolation of CD63-exposing EVs, to the best of my knowledge nobody working in the EV field has any positive experience so far with capturing beads for EVs. Capturing leads to subselection, and isolated EVs are a tiny fraction usually of all EVs present. Thus, the “EVs” shown in this TEM image are likely not representative for the (total) population of EVs as shown in the other experiments throughout this manuscript. Thus, it is essential to show representative TEM pictures of (all and thus not only CD63-selected) EVs present in the CM, because otherwise data are incomparable between experiments, and make interpretation impossible.

RESPONSE: We agreed with the reviewer that calcium in the culture media together with phosphate (in the used PBS) is insoluble (calcium phosphate crystals being formed). Therefore, we improved the sEV isolation protocol to remove cell debris, apoptotic bodies, non-vesicular particles and proteins/protein aggregates by differential centrifugation and density gradients (Response Fig12A). Before density gradients, we do not use PBS to avoid insoluble calcium phosphate crystals.

We appreciate the reviewer for pointing out the need to show representative TEM pictures of (all and thus not only CD63-selected) EVs present in the CM. Thus, we isolated sEVs by differential centrifugation (Response Fig12A) and observed sEVs by transmission electron microscopy (Response Fig14) as we mentioned in methodology by red. Meanwhile, we repeated other sEVs-related experiments throughout this manuscript to make the data comparable (Revised Fig2, Fig4, Fig5, Fig6, FigS3, FigS4, FigS5, FigS6 and FigS7). We hope that these new experiments can make the interpretation possible.

REVIEWER2, COMMENT 8: For in vitro treatment, exosomes are quantified based on measuring the concentration of protein. However, most of the measured protein –due to the crude concentration method applied (ultracentrifugation)- is likely to be from contaminants rather than from the EVs themselves.

RESPONSE: We agree that the previous ultracentrifugation method utilized in the initial manuscript was not ideal and thank you for giving us the opportunity to improve our manuscript. To get pure sEVs, we now used an improved differential centrifugation and sucrose gradient centrifugation to isolate and characterize sEVs of primary hepatocyte conditioned media according to previously published protocol (Chiou and Ansel, 2016) (Revised FigS3A). The nanoparticle tracking analysis (NTA) and electron microscopy results show that we successfully isolate sEVs by this isolated protocol (Response Fig15F and Fig14). Also, sEVs was confirmed by the finding that they expressed the exosome-specific protein markers TSG101, syntenin1, CD63, largely in F2-F3 (Response Fig15B and E). Furthermore, histone 3 and contained calreticulin were non-detectable in the hepatocyte-derived sEVs fraction (Response Fig15C), indicating that the sEVs isolated from primary hepatocytes with this protocol contained little to no contamination with cellular debris.

REVIEWER2, COMMENT 9: The described procedure for staining with PKH67 seems

wrong, because PKH67 is well known to form micelles, which need to be separated from the EVs after labelling by density gradient centrifugation (a detailed protocol has been published by the group of Marca Wauben in Nature Protocols a few years ago). To which extent this also holds true for DiI I do not know, but recent papers (see e.g. de Rond, Clin Chem 2018) do not give much hope because all these so-called “general dyes” stain not only EVs but also lipoproteins and even proteins. Without appropriate controls, these data cannot be correctly interpreted.

RESPONSE: We acknowledge that our previous method for labeling EVs was not ideal and agree with the reviewer’s comment. Based on the reviewer’s recommendation, we purified PKH67-Green-labeled sEVs secreted by hepatocytes by density gradient centrifugation to wash out PKH67 micelles, and incubated sEVs with 3T3-L1 cells visualized with High Content Analysis-Operetta (PerkinElmer, US). After 5h incubation with the labeled sEVs, the green fluorescence was observed in the recipient cells (Response Fig16A and FigS6b video), suggesting that hepatocyte-derived sEVs can directly enter adipocyte. We also purified PKH67-Green-labeled sEVs secreted by hepatocytes transfected with cy3-labeled let-7e-5p using the same method and detect the yellow fluorescence (Response Fig16G).

Response Fig16 (Revised Fig5A and 6G). Primary hepatocyte-derived sEVs containing miRNAs enter adipocytes. (A) Primary hepatocyte sEVs uptake in 3T3-L1 adipocytes after co-culture for 5 hours. (G) sEVs containing let-7e-5p mimic-cy3 internalization by 3T3-L1 preadipocytes. Purified PKH67-labeled sEVs secreted by primary hepatocytes transfected with cy3-labeled let-7e-5p were incubated with 3T3-L1 preadipocytes grown on chamber. Fluorescent images were captured by High Content Analysis-Operetta for 5 hours.

For “general dyes” staining of DiI, we added the control group of empty liposome (FormuMax) as previously described (Jerrold M. Olefsky, Cell, 5 October 2017, Pages 372-384). The DiI was added into empty liposome and incubated for 20 min before spin washing, followed by an additional wash to remove the excess dye. We injected these four mice models with labeled hepatocyte-derived sEVs and empty liposome and found that strong hepatocyte-derived EVs signal (red fluorescence) was detected in adipose. Although there exists non-specific uptake of liposome by hepatocytes in control group, the adipose has no signal detected (Response Fig17).

Response Fig17 (Revised Fig5B). *In vivo* fluorescence imaging by hepatocyte-derived EVs. (B) EVs of primary hepatocytes from WT mice treated with chow diet were injected into WT and LKO mice fed with chow diet or HFD through the tail vein. Empty liposomes are as controls. Red fluorescence levels in the liver, iWAT, and eWAT were acquired after 6 hours by IVIS spectrum.

Reviewer #3:

REVIEWER3, COMMENT 1: This study hypothesizes that the liver secretes miRNAs in exosomes and communicates to the AT regulating adipocyte metabolism. The authors demonstrate that the geranylgeranylation of hepatic proteins influences exosome biogenesis and avoid the communication between the liver and AT via exosomes. While the concept is potentially interesting, there are key experiments to support this model including:

RESPONSE: We greatly appreciate the reviewer's careful consideration in evaluating our manuscript and agree that our concept is potentially interesting to the readership of *Nature Communications*. We have responded to each comment in a point-by-point response below and hope that the new data and revised text address the concerns.

1) Demonstration of the direct transfer between the liver and AT by measuring the expression of liver specific miRNAs (miR-122) in the adipose tissue.

RESPONSE: We appreciate the reviewer's kind suggestion. Based on the reviewer's recommendation, we measured the expression of liver specific miRNAs (miR-122) in the adipose tissue of mice treated with acute/chronic HFD or EVs and found that miR-122 level was increased in adipose after HFD or HFD-EVs exposure (Response Fig18). With the validation that the labeled hepatic sEVs can enter adipocyte *in vitro* (Response Fig16) and *in vivo* (Response Fig17), we hope the reviewer will agree that hepatic sEVs can directly transfer from liver to adipose tissue *in vivo*.

Response Fig18 (Revised FigS3H, S3J and S6C). miR-122 level of adipose in mice after lipid overload. (A) miR-122 levels in mouse adipose after 6 hours of HFD treatment. (B) miR-122 levels in adipose from mice fed with chow diet and HFD at 8 weeks of age for 12 weeks. Each group was loaded with 10 μ g protein. (C) miR-122 levels in adipose from mice treated with Chow-EV and HFD-EV.

2) Analysis that the transfer of miRNAs regulates adipocyte gene expression *in vivo*. Then authors should pull down the Ago complex and assess the relative enrichment of hepatic derived miRNAs and the mRNAs target within the RISC complex.

RESPONSE: We appreciate the reviewer's suggestion to demonstrate that the transfer of miRNAs regulates adipocyte gene expression *in vivo*. To analyze whether Ago interacts with hepatocyte-derived EVs let-7e-5p to directly targets Pgc1 α in endogenous let-7e-5p knockout (let-7e-5p^{-/-}) adipocytes, RNA-protein complexes were immunoprecipitated with antibodies specific to Ago (Response Fig19A) and RNA was isolated from the complex for analysis. qPCR analysis shows that hepatocyte-derived EVs with let-7e-5p overexpression led to an increase in the amount of let-7e-5p (Response Fig19B) and Pgc1 α (Response Fig19C) bound to Ago protein. These observations suggest that by complexing with Ago, hepatocyte-derived EVs let-7e-5p was transferred to adipocyte, then targets Pgc1 α to regulate lipid deposition.

Response Fig19 (Revised FigS8). Pgc1 α of adipocyte in complex with Ago and hepatocyte-derived EVs let-7e-5p. (A) Immunoprecipitation of Ago from cell lysis of let-7e-5p^{-/-} adipocyte treated with hepatocytes-derived sEVs. Negative Control Mouse IgG was used as an IP control. (B-C) RNA ChIP analyses of Ago-miRNAs complexes subjected to Ago pull down, then total RNA isolation was analyzed for the abundance of let-7e-5p (B) and Pgc1 α mRNA (C) bound with Ago protein in recipient cells after overexpression of let-7e-5p by qPCR.

3) The metabolic analysis in the Ggpps is very descriptive. The authors should perform a deep analysis of how absence of hepatic Ggpps influence metabolic parameters (metabolic cages study) and whole glucose metabolism (hyperinsulinemic/euglycemic clamps)

RESPONSE: We agree with the reviewer's assessment that metabolic parameters and glucose metabolism should be analyzed. In our published data (Chaojun Li, J Pathol, 2018, Nov; 246(3): 277-288.), we have examined the whole-body metabolic profiles of the mice after liver-specific Ggpps deletion, and the mice were housed in metabolic cages. The respiratory exchange rate (RER) [V_{CO_2} (CO₂ production)/ V_{O_2} (O₂ consumption)] (Response Fig20A), V_{O_2} (Response Fig20B) and physical activity (Response Fig20C and D) were increased in the LKO group during fat overload. We observed no difference in food intake (Response Fig20E and F).

Response Fig20. The whole-body metabolic rate in CTL and L-Ggpps KO mice fed the RC/HFD for 12 weeks. (A) Respiratory exchange ratio (RER). (B) O₂ consumption. (C, D) Physical activity. (E, F) Food intake. n = 3 for each group. Mean ±SEM. *P < 0.05, **P < 0.01, L-Ggpps KO versus CTL; # P < 0.05, ## P < 0.01, HFD (CTL) versus RC (CTL). Reused with permissions from *Li et al., Geranylgeranyl diphosphate synthase (GGPPS) regulates non-alcoholic fatty liver disease (NAFLD)-fibrosis progression by determining hepatic glucose/fatty acid preference under high-fat diet conditions, Journal of Pathology, 246(3):277-288 (2018).*

To analyze the whole glucose metabolism, glucose tolerance tests (GTT) and insulin tolerance tests (ITT) were performed in WT and LKO mice. The data show that hepatic *Ggpps* deletion improved systemic glucose tolerance and insulin sensitivity in 20-week old mice. We think that the improved systemic glucose metabolism resulted from the improved insulin signaling pathway in both liver and adipose. Also, GTT and ITT in 8-week-old WT and *Ggpps* overexpression mice by infecting Ad-*Ggpps* into liver showed that *Ggpps* inhibited systemic glucose tolerance and insulin sensitivity (Response Fig21). Because these data show that glucose metabolism is improved in the absence of hepatic *Ggpps*, we did not use hyperinsulinemic/euglycemic clamps.

Response Fig21 (Revised Figs 4E-F). The glucose metabolism in LKO Mice. (A-B) Glucose tolerance tests (GTT) (A) and insulin tolerance tests (ITT) (B) were performed in WT and LKO mice. (C-D) The protein levels of Ggpps and insulin-stimulated phosphorylation of AKT in liver (C) and iWAT (D). (E-F) The glucose tolerance tests (E) and insulin tolerance tests (F) in 8-week-old WT and GGPPS overexpression mice by infecting Ad-Ggpps into liver.

4) RNA-Seq analysis of AT isolated from mice fed a HFD.

RESPONSE: We appreciate the reviewer's suggestion to perform RNA-Seq analysis of AT isolated from mice fed a HFD. To verify our result that lipogenesis genes decreased and lipid oxidation genes increased by HFD, we searched the GEO database and found previous Microarray and RNA-seq result in white adipose tissue of mice fed with high fat diet compared

with normal chow diet (<https://www.ncbi.nlm.nih.gov/geo/query/acc.cgi?acc=GSE14363>) (David Carling, Nature Metabolism, 2019, <https://www.ncbi.nlm.nih.gov/geo/query/acc.cgi?acc=GSE120429>). We further analyzed the lipid-metabolism genes mentioned in our manuscript and found the same change of these genes as follows (Response Table1).

Table 1 The lipid oxidation and lipogenesis genes expression profile in microarray.

Type	Gene	Accession	Gene Symbol	LocusLink	WT_NC	WT_HFD	Fold change
Lipid oxidation	peroxisome proliferative activated receptor, gamma, coactivator 1 alpha	BB745167	Ppargc1	19017	71.41	28.36	-2.52
Lipid oxidation	Fas- activated serine/threonine kinase	NM_023229	Fastk	66587	1361.76	451.66	-3.02
Lipid oxidation	acyl- CoA synthetase long- chain family member 6	BC022959	Acs16	216739	52	40.88	-1.27
Lipogenesis	sterol regulatory element binding factor 1	A1326423	Srebf1	20787	1902.64	2221.36	1.17
Lipogenesis	fatty acid binding protein 4, adipocyte	BC002148	Fabp4	11770	11262.37	11378.45	1.01
Lipogenesis	lipoprotein lipase	BC003305	Lpl	16956	10683.88	10952.66	1.03

5) Overall, the authors should perform key experiments under physiological and *in vivo* conditions to define the relevance of this study.

RESPONSE: We agree with this assessment and think that our previous data and new results illustrate a novel liver-adipose interplay of cellular metabolic networks- ‘the hepatocyte-derived EVs-stimulated adipose remodeling.’ To demonstrate this relevance under physiological condition, we perform several key experiments under acute and chronic lipid overload condition *in vivo* and *in vitro* in the revised manuscript (Response Fig22). Firstly, as TG deposition is first observed in liver, then followed by adipose tissue after acute lipid overload, we found that liver responds first to acute lipid overload. Furthermore, we observed that liver ‘remote controls’ adipose hyperplasia by adipogenesis (Response Fig22A and B), may leading to a much healthier systemic metabolism to relieve the pressure by acute lipid overload, which is accordance with the phenotype that the hepatic EVs under HFD 6 hours conduct adipose hyperplasia by adipogenesis (Response Fig22C and D). However, the long-term lipid overload causes the hepatic EV-stimulated adipose hypertrophy by lipogenesis (Response Fig22C and E), resulting in detrimental metabolic phenotype. Therefore, we found that the adipocyte size increased in iWAT and eWAT with 12 weeks HFD treatment (Response Fig22F), and adipocyte number in both iWAT and eWAT did not differ between the WT and LKO mice (Response Fig22G), suggesting that long-term HFD treated sEVs enhance lipogenesis, not adipogenesis. Besides, the phenotype of liver *Ggpps* knockout mice (Revised Fig3) exhibited the decreased sEVs secretion (Revised Fig4). Mechanically, the lipid deposition in adipose of KO mice was decreased by hepatic sEVs *in vitro* and *in vivo* (Revised Fig5), suggesting the relevance that liver secretes sEVs to communicate with adipose tissue. This liver-driven adipose remodeling is dependent on the primary hepatocyte EV miRNAs *in vivo* (Revised FigS7). Overall, these *in vitro* and *in vivo* experiments under physiological acute and chronic lipid overload conditions reveals that hepatocyte-derived EVs enhance adipose hyperplasia by adipogenesis upon acute lipid overload and conduct adipose hyperplasia by lipogenesis upon chronic lipid overload (Revised Fig8). We hope that the reviewer will agree that all data provide a strong support for the relevance of hepatocyte-derived EVs and adipose under lipid overload physiological condition *in vivo*.

Response Fig22 (Revised Fig1J-K, Fig2I-K, 3I and S4D). Hepatocyte-derived EVs enhance adipose hyperplasia by adipogenesis upon acute lipid overload and conduct adipose hyperplasia by lipogenesis upon chronic lipid overload. (A) Quantification of adipocyte number of iWAT and eWAT from HFD-fed mice at different time points. (B) Expression of fatty acid transport, lipogenesis and fatty acid oxidation-related genes in the WAT of mice following HFD exposure for indicated time. (C) Hepatocyte-derived sEVs from mice treated with 6-h and 12-week Chow diet/HFD were added to 3T3-L1 preadipocytes differentiating for 8 days. Adipocytes differentiated from 3T3-L1 adipocytes at day 10 after induction were stained with Oil Red O. (D-E) Expression of adipogenesis and lipogenesis genes in 3T3-L1 preadipocytes from Fig20C. (F) Quantification of adipocyte diameters of iWAT and eWAT from WT and LKO mice treated with chow diet and HFD. Data were collected from H&E-stained sections from three individual mice, five fields per mouse, 10–15 cells per field in each group, using ImageJ software. (G) Total amount of DNA of the inguinal WAT, epididymal WAT from WT and LKO mice.

Reviewers' comments:

Reviewer #1 (Remarks to the Author):

The authors have adequately addressed the initial concerns raised.

Reviewer #2 (Remarks to the Author):

The authors have performed additional experiments as requested which confirm their initial findings. A few minor issues seem to be remaining.

The authors changed "exosomes" to "extracellular vesicles" as suggested, but the title is lacking a "s" ("extracellular vesicle" should be changed into "extracellular vesicles"). The term "sEVs" is misleading, because methods like (T)EM are not quantitative, and minor contamination with large EVs may have a large impact on reported composition (e.g. presence of a particular miRNA). Since none of the methods used can exclude the presence of e.g. 1% or so larger EVs, I suggest to completely remove the (in my view) misleading term "sEVs" throughout the manuscript. Moreover, EVs shrink due to fixation and dehydration (as required for TEM), so diameter is 10-15% underestimated.

Please indicate the lower level of detection in Fig 15 (F).
The authors changes –as suggested- to a different isolation method.

In their reply (comment 5) the authors show a single EM image (Fig 2C), which shows some cup-shaped (and thus likely) EVs (of which some seem leaky), but in the answer they mention "By electron microscopy, we now found that these EVs were morphologically heterogeneous, although the majority exhibited a cup-like structure." Given the fact that EM images may reflect operator bias, I suggest that the authors provide a bit more information here. What is "morphologically heterogeneous"? Moreover, when looking at e.g. Fig 14 (Revised Fig 2C), I see a considerable background (small white stuff), which to me looks like protein. What is this background?

Reviewer #4 (Remarks to the Author):

The authors responded the Reviewer's criticism and added elegant data to support their interesting model on how liver and fat communicate via EVs to control fat allocation upon HFD. However, I still have some concerns:

Major

1) Perhaps my main concern is that some controls have not been done to allow a fair comparison. For example, in Fig. 2F and G, how is fat accumulation and EV secretion on chow diet? This would allow us to see whether GW4869 plays a role in these phenotypes independently of the diet. A chow diet control is also missing in Fig. 6C. Also, in Fig. 5F-H, as a rescue experiment, it should have been done in parallel with the WT and KO EV should have been used as a control. Like it is, the data only reinforces that HFD-WT EV induces adipose tissue mass, but it doesn't necessarily prove that the HFD-WT EV rescue the LKO phenotype;

2) It appears to me that the DNA analysis was performed with the whole tissue and therefore it could not be used as a surrogate for adipogenesis as adipose tissue is not only composed by adipocytes and HFD is known to promote recruitment and proliferation of many cells of the stromal-vascular fraction;

3) Also, the authors distinguish adipogenesis from lipogenesis in cell culture based on DNA and gene expression. They should measure adipogenesis and lipogenesis directly;

4) In Fig. 1K, genes should also be measured at 3h and 6h post HFD;

5) In Fig. 3E, fat mass gain on HFD is similar between WT and KO. This somewhat argues against the proposed model that suggests that HFD-induced EVs promote fat accumulation in a liver Ggpps dependent manner;

Minor

- 1) Was the hepatocyte conditioned medium added directly to preadipocytes during differentiation or diluted in adipocyte differentiation medium? If diluted, state the concentration;
- 2) Insulin sensitivity doesn't seem to be affected just by looking at the ITT. If the authors calculate KITT they will probably see the same value for both groups. Blood glucose levels seem to be affected though;
- 3) Liver Ggpps appears to be controlling EV secretion and uptake by adipocytes. The authors focus on the role of Ggpps in EV secretion, but they should also propose an explanation for why adipose tissue takes up less EVs in the LKO;
- 4) Upper and lower panels in figure 5A and 6G should be labelled;
- 5) "Because miRNA-210-3p has been reported to regulate adipocyte differentiation through Wnt signaling, we focused on the role of the identified candidate let-7e-5p in the following experiment." I didn't understand the rationale here;
- 6) It appears to have something wrong with the labels in Figure 6B. What does "EV donor" and "Recipient donor" mean?
- 7) The text requires revision for typos and grammar mistakes.

Reviewer Comments:

Reviewer #1:

REVIEWER1, COMMENT: The authors have adequately addressed the initial concerns raised.

RESPONSE: We appreciate the significant efforts of the reviewer on our work. Thank you for giving our work greater impact. THANKS!

Reviewer #2:

REVIEWER2, COMMENT 1:The authors have performed additional experiments as requested which confirm their initial findings. A few minor issues seem to be remaining.

The authors changed “exosomes” to “extracellular vesicles” as suggested, but the title is lacking a “s” (“extracellular vesicle” should be changed into “extracellular vesicles”). The term “sEVs” is misleading, because methods like (T)EM are not quantitative, and minor contamination with large EVs may have a large impact on reported composition (e.g. presence of a particular miRNA). Since none of the methods used can exclude the presence of e.g. 1% or so larger EVs, I suggest to completely remove the (in my view) misleading term “sEVs” throughout the manuscript. Moreover, EVs shrink due to fixation and dehydration (as required for TEM), so diameter is 10-15% underestimated.

RESPONSE: We apologize for this imprecise term and we have corrected the term “extracellular vesicle” to “extracellular vesicles” in title, the misleading term “sEVs” to “EVs” as highlighted by red throughout the manuscript, as the reviewer pointed out. We agree that the diameter of EV might be underestimated due to fixation and dehydration, so we have discussed that in Page 8, line 158.

REVIEWER2, COMMENT 2: Please indicate the lower level of detection in Fig 15 (F). The authors changes –as suggested- to a different isolation method.

RESPONSE: We thank the reviewer for pointing out the need to indicate the lower level of detection of Particle Metrix ZetaView® Nanoparticle Tracking Analyzer. Based on the Peak Analysis (Concentration) of [attachment redacted] provided by the company, the lower level detection of diameter can be 1 nm that we have added in the methods part (Page 29, line 597). Therefore, we believe this new equipment with detection range from 1 nm to 1000 nm diameter is a reliable tool which can detect the particles with 5.8 nm diameter

[attachment redacted].

REVIEWER2, COMMENT 3: In their reply (comment 5) the authors show a single EM image (Fig 2C), which shows some cup-shaped (and thus likely) EVs (of which some seem leaky), but in the answer they mention “By electron microscopy, we now found that these EVs were morphologically heterogeneous, although the majority exhibited a cup-like structure.” Given the fact that EM images may reflect operator bias, I suggest that the authors provide a bit more information here. What is “morphologically heterogeneous”? Moreover, when looking at e.g. Fig 14 (Revised Fig 2C), I see a considerable background (small white stuff), which too me looks like protein. What is this background?

RESPONSE: We regret the lack of a clear description and agree that the EM images may reflect operator bias. The raw EM images (Response Fig. 1) showed the majority EVs exhibited a cup-like structure, but some seemed oval-shaped, irregular shape and leaky as the reviewer suggested. We cannot exclude the possibility that EVs are deformed by shrinking due to fixation and dehydration. Thus, we correct the imprecise term “morphological heterogeneous” in our manuscript.

Based on the new differential centrifugation and density gradients method (Response Fig. 2), we have washed out proteins from EVs with PBS containing phosphate. Therefore, the background (small white stuff) might be phosphate crystals.

Response Fig. 1 Primary hepatocyte-derived EVs. (C)Electron microscopy analysis of EVs secreted by primary hepatocytes from mice fed a normal chow diet. Scale bar, 200 nm.

a

Response Fig. 2 (Revised Supplementary Fig. 3a). Isolation schematic of hepatocyte-derived EVs. (a) EVs isolation schematic.

Reviewer #4:

REVIEWER4, COMMENT 1: The authors responded the Reviewer’s criticism and added elegant data to support their interesting model on how liver and fat communicate via EVs to control fat allocation upon HFD. However, I still have some concerns: Perhaps my main concern is that some controls have not been done to allow a fair comparison. For example, in Fig. 2F and G, how is fat accumulation and EV secretion on chow diet? This would allow us to see whether GW4869 plays a role in these phenotypes independently of the diet. A chow diet control is also missing in Fig. 6C. Also, in Fig. 5F-H, as a rescue experiment, it should have been done in parallel with the WT and KO EV should have been used as a control. Like it is, the data only reinforces that HFD-WT EV induces adipose tissue mass, but it doesn’t necessarily prove that the HFD-WT EV rescue the LKO phenotype;

RESPONSE: We apologize for the lack of suitable controls. Based on the reviewer’s suggestion, we have added the control as indicated. We found that the fat accumulation and EV secretion were both decreased by GW4869 treatment (Response Fig. 3), indicating GW4869 plays a role in these phenotypes under both normal chow diet and HFD. We have revised this in Page 9, Line 178-181 ‘prior addition of GW4869 (an inhibitor of EV secretion) to the hepatocytes blocked EV production, the delivery of miR-122 from hepatocytes into 3T3-L1 adipocytes and lipid accumulation in adipocytes’. Actually, Ggpps deletion also cause the decrease of iWAT and eWAT even under chow diet treatment (Revised Fig. 3f), which suggested that both chow diet and HFD treatments could induce liver to send signals to

adipose tissue in a Ggpps dependent manner. But obviously, the signal profile should be different.

The control missed in Fig. 6c (Response Fig. 4) and Fig. 5f-h (Response Fig. 5) has also been added. We added EVs of hepatocytes from chow diet-fed mice as control to further validate our conclusion that HFD-EV could increase lipid accumulation by let-7e-5p and miR-210-3p (Response Fig. 4). We injected HFD-LKO mice with HFD-WT EV and HFD-LKO EV, the decreased adipose mass can be rescued by HFD-WT EV, but not by HFD-LKO EV (Response Fig. 5), which we have discussed in Page 14, Line 279-281 “the decreased adipose mass of HFD-fed LKO mice can be rescued by injecting hepatocyte-derived EVs from HFD-fed WT mice (HFD-WT EV) but not from HFD-fed LKO mice (HFD-LKO EV) (Fig. 5f)”.

Response Fig. 3 (Revised Fig. 2f-g). Hepatocytes Remodel Adipocytes via EVs. (f) Hepatocyte-derived EVs (Chow-EV or HFD-EV) were added to 2-day differentiated 3T3-L1 preadipocytes. GW4869 (10 μ M) was used to treat donor WT cells to inhibit EV formation. Adipocytes that differentiated from 3T3-L1 adipocytes at day 6 after MDI induction were stained with ORO. (g) EV production by primary hepatocytes treated with 10 μ M GW4869.

Response Fig. 4 (Revised Fig. 6c). The Oil Red O staining of adipocytes treated by hepatic EVs. (C) EVs from WT primary hepatocytes of Chow or HFD-fed mice transfected with miRNAs inhibitors (Chow-EV, HFD-EV, HFD-EV-WT, HFD-EV-miR-210-3p^{-/-}, HFD-EV-let-7e-5p^{-/-}, HFD-EV-miR-31-5p^{-/-}) were added to 3T3-L1

preadipocytes differentiating for 2 days. Adipocytes that differentiated from 3T3-L1 adipocytes at day 8 after MDI induction were stained with Oil Red O.

Response Fig. 5 (Revised Fig. 5f-h). Adipose remodelling by Hepatocyte-derived EVs. (f) EVs produced by primary hepatocytes isolated from LKO or WT mice fed an HFD (HFD-LKO EV or HFD-WT EV) were injected via tail vein every 3 days for 3 weeks into 8-week-old HFD-LKO mice. The mass of iWAT and eWAT in HFD-LKO mice was compared, as was the fat index of iWAT and eWAT. Empty liposomes were used as controls. (g-h) Expression of genes related to lipogenesis and lipolysis in iWAT (g) and eWAT (h) of the mice from (f).

REVIEWER4, COMMENT 2: It appears to me that the DNA analysis was performed with the whole tissue and therefore it could not be used as a surrogate for adipogenesis as adipose tissue is not only composed by adipocytes and HFD is known to promote recruitment and proliferation of many cells of the stromal-vascular fraction;

RESPONSE: We acknowledge that the total DNA analysis was not appropriate for measuring adipogenesis because of the involvement of other cells in SVF, like macrophages. To address this, we digested WAT and separated into mature adipocytes and SVF for DNA analysis. The results showed the numbers of both SVF and mature adipocytes in iWAT but not eWAT increased by 1-week HFD (Response Fig. 6), suggesting HFD can stimulate adipogenesis. We have discussed this in Page 6, Line 119-121 “the adipocyte number increased at 1 week as measured by DNA of total adipose tissue, mature adipocytes and the stromal vascular fraction (SVF)”.

Response Fig. 6 (Revised Supplementary Fig. 1g-h). The number of mature adipocytes and the SVF in WAT. (g-h) Quantification of the number of mature adipocytes and the SVF in iWAT and eWAT from HFD-fed mice at the indicated time points.

REVIEWER4, COMMENT 3: Also, the authors distinguish adipogenesis from lipogenesis in cell culture based on DNA and gene expression. They should measure adipogenesis and lipogenesis directly;

RESPONSE: We regret for not measuring adipogenesis and lipogenesis directly. To determine whether EVs stimulated adipogenesis in 3T3-L1 preadipocytes, the 5-ethynyl-2'-deoxyuridine (EdU) incorporation assay was performed. After challenging with EVs from 6-hour-HFD or 12-week-HFD treated mice for 1 day, 3T3-L1 preadipocytes was stained by EdU. The statistic of positive fluorescence results indicated there was no significant change of adipogenesis (Response Fig. 7). For lipogenesis assay, the lipid accumulation of 3T3-L1 adipocytes were measured by a colorimetric assay (Cayman Chemical Company, MI, USA). The relative lipid staining results suggested there was no significant change of lipogenesis by HFD-6h EVs, while HFD-12w EVs stimulated lipogenesis (Response Fig. 7). Taken together, these new results are in accordance with our previous findings that the effects of EVs on adipogenesis and lipogenesis are different. We have discussed the new results with EdU Click Assay and lipogenesis assay in Page9, line 186-188 with Supplementary Fig. 3m-n and Methods (Page 31-32, line 643-647).

Response Fig. 7 (Revised Supplementary Fig. 3m-n). Measurement of adipogenesis and lipogenesis in adipocytes. (m) Quantification of EdU-positive areas in 3T3-L1 preadipocytes from Fig. 2i. **(n)** Quantification of lipid content in 3T3-L1 preadipocytes from Fig. 2i.

REVIEWER4, COMMENT 4: In Fig. 1K, genes should also be measured at 3h and 6h post HFD;

RESPONSE: We thank the reviewer for pointing out the need to measure the adipogenesis genes at 3h and 6h. We found there were no significant change of these genes at 3h and 6h (Response Fig. 8), which is in accordance with previous findings that these genes start to express after 18–24 h adipocyte differentiation in sequence (Tang QQ, Lane MD. Annu Rev Biochem. 2012;81:715-36).

Response Fig. 8 (Revised Supplementary Fig .1j). Adipogenesis genes in WAT of mice by HFD exposure. (j) Expression of genes related to adipogenesis in the WATs of HFD-fed mice at the indicated times.

REVIEWER4, COMMENT 5: In Fig. 3E, fat mass gain on HFD is similar between WT and KO. This somewhat argues against the proposed model that suggests that HFD-induced EVs promote fat accumulation in a liver Ggpps dependent manner;

RESPONSE: We agree with the reviewer and also find this phenotype between WT and KO mice. Several reasons may account for this phenotype. Firstly, the short-term lipid overload caused hepatic EV-stimulated adipose hyperplasia (Revised Fig. 1-2), while long-term lipid overload induced adipose hypertrophy (Revised Fig. 3). This short-term phenomenon might occur because, from an anatomical perspective, the liver accesses consumed nutrients more easily than adipose tissue or skeletal muscle and the acute influence on adipose by HFD may mainly due to liver. However, under long-term HFD (12 weeks) treatment (Revised Fig. 3), adipose has also been influenced by long-term dietary fatty acids digested from intestine, even other endocrine organ, not only liver. Therefore, the similar fat mass gain on long-term HFD between WT and LKO in Fig. 3e may result from adipose itself directly.

Secondly, both normal chow diet and HFD could regulate adipose tissue in a liver-Ggpps dependent manner, but the hepatic signal evoked by diets might be different. As *in vitro* experiment shown in adipocytes (Response Fig. 3), hepatic EVs participate in liver-adipose axis under both chow diet and HFD. Based on the previous findings that hepatic EVs from short-term HFD or long-term HFD treatment mice had different function on adipogenesis and lipogenesis (Revised Fig .2i-k), we think the container profiles in hepatic EVs are also different under chow diet and HFD condition. This distinction may due to the different expression level of liver Ggpps under chow diet and HFD, which we will further investigate. On the other hand, *Ggpps* deficiency in liver may influence some hepatokines and metabolites secretion, like bile acids and GGPP (synthesized by GGPPS from its substrate FPP), which are previously shown to remodel adipose (Schrauwen P, Cell Metab. 2015 Sep 1;22(3):418-26; Weivoda MM, Hohl RJ. Bone. 2012 Feb;50(2):467-76.). Thus, the adipose under HFD treatment in LKO mice are different from that in WT mice because of the specific hepatic EV containers and other metabolites regulated by different liver Ggpps expression

under chow diet and HFD. Therefore, this similar fat mass gain phenotype may also due to the effect of liver indirectly.

In our manuscript, dietary fatty acids induced *Ggpps* expression in liver to regulate fat accumulation via EVs postprandially, especially under HFD. Therefore, liver fails to govern adipose lipid deposition in liver-specific *Ggpps* deficiency mice (Revised Fig. 3). The proposed model that hepatic EVs govern adipose remodeling in a liver *Ggpps* dependent manner is also apply to chow diet, similar with postprandial condition with different signal from liver. Here, we emphasized the role of liver under HFD-induced fat accumulation, proposing the model that the liver first senses HFD and then orchestrate liver-adipose communication to remodel adipose by HFD-induced EVs in a liver *Ggpps* dependent manner not only in short-term lipid overload but also in long-term lipid overload. We have discussed this phenotype in Page 23, line 459-462.

REVIEWER4, COMMENT 6: Was the hepatocyte conditioned medium added directly to preadipocytes during differentiation or in adipocyte differentiation medium? If diluted, state the concentration;

RESPONSE: We apologize for these vague statements. As the reviewer has correctly pointed out, we added the concentration statement “To induce the differentiation of 3T3-L1 preadipocytes, hepatocyte-conditioned medium was diluted 50-fold.” to Methods (Revised Page 26, line 528-529) as highlighted by red.

REVIEWER4, COMMENT 7: Insulin sensitivity doesn't seem to be affected just by looking at the ITT. If the authors calculate kITT they will probably see the same value for both groups. Blood glucose levels seem to be affected though;

RESPONSE: We apologize for this imprecise sentence. We calculated kITT (100 multiplied value of coefficient calculated using linear regression with single variable, with time(unit:minute) as x axis and the natural logarithm of glucose level(unit:mmol/L) as y axis), and found there was no significant change between WT and KO mice under chow diet with $p=0.62$ (Response Fig. 9). However, the glucose level was truly affected. Therefore, we correct this sentence to “hepatic *Ggpps* deletion improved systemic glucose tolerance but not insulin sensitivity (Supplementary Fig. 4e and f).” (Revised Page 11, line 217-218) as highlighted by red. We hope that this is a clearer statement.

Response Fig. 9 kITT of WT and LKO mice.

REVIEWER4, COMMENT 8: Liver *Ggpps* appears to be controlling EV secretion and uptake by adipocytes. The authors focus on the role of *Ggpps* in EV secretion, but they should also propose an explanation for why adipose tissue takes up less EVs in the LKO;

RESPONSE: We thank the reviewer for figuring out the phenomenon that adipose tissue takes up less EVs in the LKO mice. Since the adipose has been remodeled in LKO mice as we shown in Revised Fig. 3, we think liver-specific *Ggpps* deficiency influences not only the hepatic EV secretion but also adipose EV uptake. *Ggpps* deficiency in liver may influence some hepatokines and metabolites secretion, like bile acids and GGPP (synthesized by GGPPS from its substrate FPP), which are previously shown to influence adipose browning (Schrauwen P, Cell Metab. 2015 Sep 1;22(3):418-26.) and adipogenesis (Weivoda MM, Hohl RJ. Bone. 2012 Feb;50(2):467-76.). Therefore, the corresponding change of adipose in the LKO mice may also due to these secretory factors, which cause itself taking up less hepatic EVs to adapt new physiological status. We have discussed this phenotype in Page 23, line 462-466.

REVIEWER4, COMMENT 9: Upper and lower panels in figure 5A and 6G should be labelled;

RESPONSE: We regret the lack of time labels (0-5h) in the initial manuscript. We now provided new figures with the white time labels in the revised manuscript (Response Fig. 10).

Response Fig. 10 (Revised Fig. 5a and 6g). *In vivo* fluorescence imaging by hepatocyte-derived EVs. (a) Primary hepatocyte EV uptake in 3T3-L1 adipocytes after 5 hours of co-culture. (g) Purified PKH67-labelled EVs secreted by primary hepatocytes transfected with Cy3-labelled let-7e-5p were incubated with 3T3-L1 preadipocytes cultured in a chamber. Fluorescence images were captured with an Operetta High Content Imaging system for 5 hours.

REVIEWER4, COMMENT 10: ‘Because miRNA-210-3p has been reported to regulate adipocyte differentiation through Wnt signaling, we focused on the role of the identified candidate let-7e-5p in the following experiment.’ I didn’t understand the rationale here;

RESPONSE: We apologize for this confusing sentence, so we change this statement to ‘which is accordance with previous finding that miRNA-210-3p regulate adipocyte differentiation through Wnt signaling’ in the revised manuscript as highlighted by red (Revised Page 16, line 317-318).

REVIEWER4, COMMENT 11: It appears to have something wrong with the labels in Figure 6B. What does ‘EV donor’ and ‘Recipient donor’ mean?

RESPONSE: We apologize for the unclear description about the ‘EV donor’ and ‘Recipient donor’ in the initial manuscript. Thus, we change ‘EV donor’ to ‘Hepatocyte’, and ‘Recipient donor’ to ‘Adipocyte’ in the revised Fig.6b (Response Fig. 11).

Response Fig. 11 (Revised Fig. 6b). Primary Hepatocyte EVs MiRNAs in Adipocyte. (b) The relative miRNAs levels in miRNA knockout adipocytes that received WT, miR122^{-/-}, let-7e-5p^{-/-}, miR-31-5p^{-/-}, or miR-210-3p^{-/-} EVs derived from primary hepatocytes. Data were normalized to U6 levels.

REVIEWER4, COMMENT 12: The text requires revision for typos and grammar mistakes.

RESPONSE: We apologize for the typos and grammar mistakes. We have asked SPRINGER NATURE Author Services to correct all the mistakes in our revised manuscript as editing certification shown below. The paper was edited for grammar, phrasing, and punctuation. In addition, many edits were made to further improve the flow and readability of the text. We believe our manuscript has been greatly strengthened to reach the quality of *Nature Communications*.

This document certifies that the manuscript

Liver Governs Adipose Remodeling Through Extracellular Vesicles in Response to Lipid Overload

prepared by the authors

Yue Zhao, Meng-Fei Zhao, Shan Jiang, Jing Wu, Jia Liu, Xian-Wen Yuan, Di Shen, Jing-Zi Zhang, Nan Zhou, Jian He, Lei Fang, Xi-Tai Sun, Bin Xue,...

was edited for proper English language, grammar, punctuation, spelling, and overall style by one or more of the highly qualified native English speaking editors at SNAS.

This certificate was issued on **August 28, 2019** and may be verified on the SNAS website using the verification code **C289-D3B8-0715-CC01-345P**.

Neither the research content nor the authors' intentions were altered in any way during the editing process. Documents receiving this certification should be English-ready for publication; however, the author has the ability to accept or reject our suggestions and changes. To verify the final SNAS edited version, please visit our verification page at secure.authorservices.springernature.com/certificate/verify. If you have any questions or concerns about this edited document, please contact SNAS at support@as.springernature.com.

SNAS provides a range of editing, translation, and manuscript services for researchers and publishers around the world. For more information about our company, services, and partner discounts, please visit authorservices.springernature.com.

Reviewers' comments:

Reviewer #2 (Remarks to the Author):

The authors have dealt with all my concerns, except their answer to question 2, which is incorrect.

I just checked with a few (bio) medical engineers who have ample experience in small particle detection by NTA, and both came up with 50-70 nm. Please double check and correct, or explain better. A lower limit of detection of 1 nm by an optical method simply doesn't make sense. Alternatively, physical terms like resolution etc. are being mixed up (hence the 1 nm), so please check.

Reviewer #4 (Remarks to the Author):

The authors addressed most of my concerns, but there are minor issues remaining.

- 1) "Meanwhile, the decreased adipose mass of HFD-fed LKO mice can be rescued by injecting hepatocyte-derived EVs from HFD-fed WT mice (HFD-WT EV) but not from HFD-fed LKO mice (HFD-LKO EV)" – as mentioned before, one cannot use the word rescue if the WT is not included. The authors can only say that HFD-WT EV increases fat accumulation in the LKO mice while HFD-LKO EV does not.
- 2) In Supplementary Fig. 1g-h it says "Relative DNA levels" in the Y axis. I assume this is relative to control, right? More importantly, is this reflecting total DNA content per fat pad? This should be indicated in the methods and/or figure.
- 3) The authors use EdU and infer about adipocyte differentiation, but EdU marks proliferation, not adipogenesis. There might be more adipocytes in culture due to increased differentiation but this doesn't necessarily associate with more proliferation. They should state that there is no difference in proliferation and not adipogenesis. Having said that, I am convinced by the expression of aP2, Pparg and Adipoq that adipogenesis is not different.
- 4) Also, they measure lipid accumulation and infer about lipogenesis. Lipogenesis is measured by lipid synthesis. Lipid accumulation reflects the balance of lipid synthesis (lipogenesis) and utilization (lipolysis and oxidation). Therefore, the authors should state that they measured lipid accumulation, not lipogenesis, unless they do measure lipid synthesis.
- 5) "Thus, the similar fat mass gain on HFD between WT and LKO mice (Fig. 3e) may due to adipose itself influenced by long-term HFD directly or the corresponding adipose remodelling by the different hepatic EVs under chow diet or HFD indirectly. More interestingly, we have found adipose takes up less EVs in the liver Ggpps deficiency mice, which may result from decreased Ggpps-dependent hepatokines and metabolites secretion, like bile acids and GGPP (synthesized by GGPPS from its substrate FPP), previously shown to influence adipose remodeling." - this part is confusing. I am not convinced by the explanation why WT and LKO have similar levels of weight gain on HFD. The data tells me that HFD leads to increased adiposity in a liver Ggpps independent manner, while liver Ggpps controls adiposity both in chow diet and HFD. I buy the notion that liver Ggpps controls postprandial remodeling of adipose tissue via EVs, as demonstrated in other experiments, but my previous point was that the authors cannot say that this phenomenon is restricted to HFD. I am satisfied if they just don't state that. It is hard to understand what they mean by "adipose itself influenced by long-term HFD directly" or "different hepatic EVs under chow diet or HFD indirectly". Finally, the explanation why adipose takes up less EVs in liver Ggpps deficiency mice is not convincing. Have the proposed molecules been shown to control EV uptake? If yes, mention it. If not, I would not speculate. I would just highlight the observation and mention that it is not known why that might happen.

Reviewer Comments:

Reviewer #2:

REVIEWER2, COMMENT: The authors have dealt with all my concerns, except their answer to question 2, which is incorrect. I just checked with a few (bio) medical engineers who have ample experience in small particle detection by NTA, and both came up with 50-70 nm. Please double check and correct, or explain better. A lower limit of detection of 1 nm by an optical method simply doesn't make sense. Alternatively, physical terms like resolution etc. are being mixed up (hence the 1 nm), so please check.

RESPONSE: We appreciate the check of the reviewer on small particle detection by NTA. We apologize for this incorrect description and asked the engineer of Particle Metrix ZetaView® Nanoparticle Tracking Analyzer again to ensure the lower limit of detection. The engineer told us the lower limit detection is around 50 nm, so we have corrected it in the methods part as highlighted by red (Page 29, line 587).

Reviewer #4:

REVIEWER4, COMMENT 1: The authors addressed most of my concerns, but there are minor issues remaining.

“Meanwhile, the decreased adipose mass of HFD-fed LKO mice can be rescued by injecting hepatocyte-derived EVs from HFD-fed WT mice (HFD-WT EV) but not from HFD-fed LKO mice (HFD-LKO EV)” – as mentioned before, one cannot use the word rescue if the WT is not included. The authors can only say that HFD-WT EV increases fat accumulation in the LKO mice while HFD-LKO EV does not.

RESPONSE: We apologize for this imprecise description and we have corrected this statement to “HFD-WT EV increases fat accumulation in the LKO mice while HFD-LKO EV does not”, as the reviewer pointed out. We highlighted this statement in Page 14, line 276-277.

REVIEWER4, COMMENT 2: In Supplementary Fig. 1g-h it says “Relative DNA levels” in the Y axis. I assume this is relative to control, right? More importantly, is this reflecting total DNA content per fat pad? This should be indicated in the methods and/or figure.

RESPONSE: We thank the reviewer for pointing out the need to indicate the detail of this quantitative analysis. The “Relative DNA levels” in the Y axis in Supplementary Fig. 1g-h means relative to control as you think. Based on the “Mature adipocytes and SVF isolation” method, there was somehow loss of cells according to the degree of digestion. To make sure the consistency of results, we prepared the samples at the same time. Therefore, the data

reflect the relative total DNA content per fat pad as we highlighted by red in figure legend of Supplementary Fig. 1g-h.

REVIEWER4, COMMENT 3: The authors use EdU and infer about adipocyte differentiation, but Edu marks proliferation, not adipogenesis. There might be more adipocytes in culture due to increased differentiation but this doesn't necessarily associate with more proliferation. They should state that there is no difference in proliferation and not adipogenesis. Having said that, I am convinced by the expression of aP2, Pparg and Adipoq that adipogenesis is not different.

RESPONSE: We apologize for the incorrect description and agree that EdU marks proliferation, not adipogenesis. Thus, we correct the incorrect statement to "The 5-ethynyl-2'-deoxyuridine (EdU)-positive cells showed there is no difference in proliferation between HFD-6h EV treated cells and HFD-12w EV treated cells (Supplementary Fig. 3m)." as highlighted by red in Page 9, line 182-184.

REVIEWER4, COMMENT 4: Also, they measure lipid accumulation and infer about lipogenesis. Lipogenesis is measured by lipid synthesis. Lipid accumulation reflects the balance of lipid synthesis (lipogenesis) and utilization (lipolysis and oxidation). Therefore, the authors should state that they measured lipid accumulation, not lipogenesis, unless they do measure lipid synthesis.

RESPONSE: We are sorry for this imprecise statement. Based on the reviewer's suggestion, we have changed 'lipogenesis' to 'lipid accumulation' in the revised manuscript as highlighted by red (Revised Page 9, line 186; Page 10, line 189).

REVIEWER4, COMMENT 5: " Thus, the similar fat mass gain on HFD between WT and LKO mice (Fig. 3e) may due to adipose itself influenced by long-term HFD directly or the corresponding adipose remodelling by the different hepatic EVs under chow diet or HFD indirectly. More interestingly, we have found adipose takes up less EVs in the liver Ggpps deficiency mice, which may result from decreased Ggpps-dependent hepatokines and metabolites secretion, like bile acids and GGPP (synthesized by GGPPS from its substrate FPP), previously shown to influence adipose remodeling." - this part is confusing. I am not convinced by the explanation why WT and LKO have similar levels of weight gain on HFD. The data tells me that HFD leads to increased adiposity in a liver Ggpps independent manner, while liver Ggpps controls adiposity both in chow diet and HFD. I buy the notion that liver Ggpps controls postprandial remodeling of adipose tissue via EVs, as demonstrated in other experiments, but my previous point was that the authors cannot say that this phenomenon is restricted to HFD. I am satisfied if they just don't state that. It is hard to understand what they mean by "adipose itself influenced by long-term HFD directly" or " different hepatic EVs under chow diet or HFD indirectly". Finally, the explanation why adipose takes up less EVs in liver Ggpps deficiency mice is not convincing. Have the proposed molecules been shown

to control EV uptake? If yes, mention it. If not, I would not speculate. I would just highlight the observation and mention that it is not known why that might happen.

RESPONSE: We regret for misunderstanding the reviewer's previous point. Actually, we agree that this phenomenon is not restricted to HFD as the reviewer points out. The proposed model that hepatic EVs govern adipose remodeling in a liver Ggpps dependent manner is also apply to chow diet as indicated by Fig.2f and Fig. 3f. Therefore, we deleted the statement in the previous discussion (Page 22, line 14-17).

We are sorry for the speculation about EV uptake by adipose, which we have deleted in the manuscript. We only mention the observation and address that it deserved further study to unravel molecular mechanism in Page 22, line 454-456.

REVIEWERS' COMMENTS:

Reviewer #4 (Remarks to the Author):

Related to my previous concerns, in the abstract the authors should correct the following sentences:

- 1) "Lipid overload fails to govern adipose lipid deposition in liver-specific Ggpps deficiency mice." It actually still does increase adipose deposition to some extent, although the baseline fat accumulation in the LKO mouse is lower. I would change this sentence to something like: "Consistently, liver-specific Ggpps deficiency mice have reduced fat adipose deposition"
- 2) "...and these miRNAs enhance adipocyte lipid accumulation by increasing lipogenesis and inhibiting lipid oxidation". I would remove "by increasing lipogenesis and inhibiting lipid oxidation" since this has been mostly inferred by gene expression.

There are also typos and grammar mistakes in several places of the MS, such as in lines 61 ("organs within the tissue"?), 77 ("This phenomenon became apparent" or inhibited?), 190 ("that the" should read "the"), 251 (mediate not "mediates"), 272 (change not changed), 446 (adipocyte- not "adipocytes-").

Dear Reviewer 4,

Thank you for the insightful and constructive comments on our manuscript entitled ‘Liver Governs Adipose Remodelling via Extracellular Vesicles in Response to Lipid Overload’ (NCOMMS-18-34647C). We have revised our manuscript according to the comments raised. Please find the point-by-point responses to comments below:

Reviewer Comments:

Reviewer #4:

Related to my previous concerns, in the abstract the authors should correct the following sentences:

1)"Lipid overload fails to govern adipose lipid deposition in liver-specific Ggpps deficiency mice." It actually still does increase adipose deposition to some extent, although the baseline fat accumulation in the LKO mouse is lower. I would change this sentence to something like: "Consistently, liver-specific Ggpps deficiency mice have reduced fat adipose deposition"

RESPONSE: We appreciate the check of the reviewer on the abstract. Now, we have changed the sentence to “Consistently, liver-specific Ggpps deficiency mice have reduced fat adipose deposition” in Page 2, Line 35-36 by tracked change.

2) "...and these miRNAs enhance adipocyte lipid accumulation by increasing lipogenesis and inhibiting lipid oxidation". I would remove "by increasing lipogenesis and inhibiting lipid oxidation" since this has been mostly inferred by gene expression.

RESPONSE: According to the suggestion of the reviewer, we have removed “by increasing lipogenesis and inhibiting lipid oxidation” in Page 2, Line 39 by tracked change.

There are also typos and grammar mistakes in several places of the MS, such as in lines 61 ("organs within the tissue"?), 77 ("This phenomenon became apparent" or inhibited?), 190 ("that the" should read "the"), 251 (mediate not "mediates"), 272 (change not changed), 446 (adipocyte- not "adipocytes-").

RESPONSE: We apologize for typos and grammar mistakes. We have checked the manuscript again and corrected these mistakes by tracked change in the revised manuscript (Page 3, Line 71; Page 4, Line 87; Page 9, Line 212; Page 12, Line 274; Page 13, Line 296; Page 21, Line 479).